# Is L-Glutamate Toxic to Neurons and Thereby Contributes to Neuronal Loss and Neurodegeneration? A Systematic Review

**DOI:** 10.3390/brainsci12050577

**Published:** 2022-04-29

**Authors:** Maryam N. AL-Nasser, Ian R. Mellor, Wayne G. Carter

**Affiliations:** 1Department of Biological Sciences, College of Science, King Faisal University, P.O. Box 400, Al-Ahsa 31982, Saudi Arabia; maryam.al-nasser1@nottingham.ac.uk; 2School of Life Sciences, Faculty of Medicine and Health Sciences, University of Nottingham, Nottingham NG7 2RD, UK; ian.mellor@nottingham.ac.uk; 3School of Medicine, Royal Derby Hospital Centre, University of Nottingham, Derby DE22 3DT, UK

**Keywords:** excitotoxicity, L-glutamate, mitochondrial dysfunction, neurodegeneration, neuroinflammation, oxidative stress, protein aggregation

## Abstract

L-glutamate (L-Glu) is a nonessential amino acid, but an extensively utilised excitatory neurotransmitter with critical roles in normal brain function. Aberrant accumulation of L-Glu has been linked to neurotoxicity and neurodegeneration. To investigate this further, we systematically reviewed the literature to evaluate the effects of L-Glu on neuronal viability linked to the pathogenesis and/or progression of neurodegenerative diseases (NDDs). A search in PubMed, Medline, Embase, and Web of Science Core Collection was conducted to retrieve studies that investigated an association between L-Glu and pathology for five NDDs: Alzheimer’s disease (AD), Parkinson’s disease (PD), multiple sclerosis (MS), amyotrophic lateral sclerosis (ALS), and Huntington’s disease (HD). Together, 4060 studies were identified, of which 71 met eligibility criteria. Despite several inadequacies, including small sample size, employment of supraphysiological concentrations, and a range of administration routes, it was concluded that exposure to L-Glu in vitro or in vivo has multiple pathogenic mechanisms that influence neuronal viability. These mechanisms include oxidative stress, reduced antioxidant defence, neuroinflammation, altered neurotransmitter levels, protein accumulations, excitotoxicity, mitochondrial dysfunction, intracellular calcium level changes, and effects on neuronal histology, cognitive function, and animal behaviour. This implies that clinical and epidemiological studies are required to assess the potential neuronal harm arising from excessive intake of exogenous L-Glu.

## 1. Introduction

Although L-glutamate (L-Glu) is a nonessential amino acid, it is the most abundant excitatory neurotransmitter in the central nervous system (CNS) [1,2,3]. It has several critical roles in brain development and functionality, including facilitating communication between neurons, and contributes to neuronal plasticity and energy supply [2,3]. Additionally, it is involved in the regulation of learning and memory and provides persistent synaptic strengthening, termed neural long-term potentiation [2,3].

During neonatal development of the CNS, L-Glu acts as a neurotrophic factor [3,4]. Three distinct compartments in the brain involve L-Glu actions: presynaptic neurons, postsynaptic neurons, and glial cells [5,6]. Notably, the glutamate–glutamine cycle is a vital process in which synaptic terminals and glial cells cooperate to maintain an adequate level of L-Glu (glutamate homeostasis) [5,6].

Within an excitatory synaptic cleft, the L-Glu concentration normally rises to a relatively high level of ≈1 mM after the arrival of an action potential at the presynaptic nerve terminal, but only remains at this concentration for a few milliseconds and, thereafter, returns to its normal nanomolar levels due to association with high-affinity transporters expressed on neurons and glia [7,8]. Generally, L-Glu is required to maintain healthy and normal brain function. However, its accumulation is linked to neurotoxicity and neurodegeneration [7,9].

Several lines of evidence have suggested that impaired L-Glu homeostasis is linked to neuronal pathology and death in neurodegenerative diseases (NDDs), such as Alzheimer’s disease (AD) [10,11], Parkinson’s disease (PD) [7,12,13], multiple sclerosis (MS) [14,15,16] amyotrophic lateral sclerosis (ALS) [17,18,19], and Huntington’s disease (HD) [20,21].

Since L-Glu has been implicated in the pathogenesis and/or progression of NDDs, there is a need to study and monitor L-Glu levels. In vitro studies have demonstrated that L-Glu induces oxidative stress by increasing intracellular reactive oxygen species (ROS), reducing levels and function of the antioxidant defence system, and impairment of mitochondrial function, ultimately leading to neuronal damage and loss [22,23,24,25,26]. Similar outcomes were observed after systemic L-Glu administration in vivo, which triggered induction of oxidative stress and histopathological alterations in brain tissues [27,28]. In addition, there are other neuropathological mechanisms induced by L-Glu, including excitotoxicity, in which L-Glu overactivates its receptors, including those for N-methyl-D-aspartate (NMDA), α-amino-3-hydroxy-5-methyl-4-isoxazolepropionic acid (AMPA), kainate acid (KA), and metabotropic glutamate (mGlu), resulting in an excessive influx of calcium ions (Ca^2+^) into neurons, leading to the generation of reactive radicals and increased levels of proapoptotic proteins [29,30,31,32]. L-Glu treatment can also trigger an accumulation of toxic proteins and impair neuronal function [33,34,35], as well as contribute to neuroinflammation [14,36].

Thus, collectively, excessive L-Glu levels can contribute to neuronal damage and degradation through different aetiology processes, including oxidative stress, excitotoxicity, mitochondrial dysfunction, inflammation, and protein aggregation, and thereby contribute to the progression of NDDs [37,38,39,40]. It therefore remains critical to brain health that the levels of L-Glu remain optimal. However, exogenous free L-Glu can be introduced to the human body through the diet, for example, as natural foods or food additives, such as monosodium L-glutamate (MSG) monohydrate, a chemical added to food to enhance its flavour [28,41,42]. Furthermore, toddlers can be exposed to MSG in milk formula, infant food that is cooked or mashed, and canned fruits and vegetables [43,44]. Vaccines could also be a source for L-Glu entry into the body, as the Centers for Disease Control and Prevention (CDC) indicates that MSG is used as a stabiliser or preservative for some vaccines [45].

In view of this concern regarding the incorporation of excessive exogenous L-Glu, and the potential for accumulation at the synaptic cleft and associated risk to neuronal survival [46,47], this systematic research was undertaken to provide a comprehensive, unbiased analysis of the effects of L-Glu on neuronal viability and implications for the pathology of NDDs, such as AD, PD, MS, ALS, and HD.

## 2. Materials and Methods

This review was conducted based on the Preferred Reporting Items for Systematic Reviews and Meta-Analyses (PRISMA) [48,49].

### 2.1. Search Scheme

A systematic electronic database search was performed from 6 to 16 October 2020 on Medline (OvidSP), Embase (OvidSP), Web of Science Core Collection, and PubMed. The aim was to identify research studies that reported the influence of L-Glu upon neuronal viability and resulting NDDs. Controlled search vocabularies (MeSH) were used and involved a combination of the following: (a) excitotoxic* amino acid OR L-glutamate OR glutamate neurotoxic* OR L-glutamic acid OR monosodium glutamate; (b) neurodegenerative diseases OR nerve degeneration OR neurodegenerat* OR neurotoxicity; (c) mitochondrial dysfunction OR protein toxicity, excitotoxicity OR oxidative damage OR inflamma* OR neuroinflammat*. Hand searching of related papers generated by bibliography screening was also conducted.

### 2.2. Data Extraction and Collection

Data were extracted from eligible articles, and information collected for the following variables: authors, year of publication, in vitro or in vivo study, dose or concentration of L-Glu, route of application, toxicity assay and method, overall study outcome, and conclusions.

### 2.3. Eligibility Criteria

All search results (n = 4060) were imported into EndNote (Clarivate Analytics), and automatic deduplication was performed. A manual check of the title and abstract screening was then undertaken to identify studies considered relevant to the prespecified inclusion criteria.

#### 2.3.1. Inclusion Criteria

All in vitro findings that were original studies directly investigating the effect of L-Glu on molecular mechanisms chiefly associated with the pathology of AD, PD, MS, ALS, or HD, and in which L-Glu influenced neuronal viability, were included. Similarly, all in vivo (animal) evidence that focused upon the direct impact of L-Glu on neuronal molecular processes that resulted in diseases, such as that observed in NDDs, specifically AD, PD, MS, ALS, or HD, was included. The first and last authors of this manuscript reviewed all papers that met the inclusion criteria and independently performed the data extraction and discussed any anomalies.

#### 2.3.2. Exclusion Criteria

Studies were excluded if they focused on L-Glu toxicity in organs other than the brain or were focused on NDDs other than those specified above, were published in a language other than English, were performed with nonhuman neurons or tissue, or were review articles, editorials, or conference abstracts.

## 3. Results

The primary database search resulted in a total of 4043 articles, and then hand searching for relevant papers added a further 17 related papers. After removing duplicates, 2467 papers were then excluded based upon the title and abstract screening. This yielded 864 articles, and these were subjected to full-text assessment. A total of 793 of these studies were then excluded based on unfulfilled predefined eligibility criteria and for the following reasons: not relevant (n = 94), review (n = 45), animal in vitro studies (n = 583), non-English language (n = 4), animal or human neurons in another organ: retina (n = 32), cochlea, heart, liver (n = 5), focus on different neurological diseases (n = 22), predictive data from a virtual experimental system (n = 1), L-Glu mixed with another compound (n = 3), modified or transgenic human neurons, human amyloid precursor protein (APP) mutation, and senescence by X-irradiation (n = 4). This resulted in a total of 71 articles that met the inclusion criteria. These results are shown as a flowchart detailing the stages of study retrieval and selection based on PRISMA (Figure 1). Of the 71 included studies, most were in vivo studies (n = 47), while 23 studies were in vitro, and only 1 study used mixed methods.

### 3.1. In Vitro Studies Evaluating L-Glu Toxicity in Human Neurons

In vitro studies were only included if they considered a direct effect of L-Glu on human neurons. A number of different neuronal models were considered: human neuroblastoma cells (SH-SY5Y) (n = 15) [51,52,53,54,55,56,57,58,59,60,61,62,63,64,65,66], SK-N-BE (n = 1) [67], IMR-32 (n = 1) [68], HCN-1A human cortical neurons (n = 2) [69,70], human embryonic stem cell (HESC) line H9 (n = 1) [71], HB1.F3 human neural stem cells (NSCs) (n = 1) [72], and primary human fetal brain tissue (14- to 18-week aborted foetuses) (n = 2) [73,74], with studies listed in chronological order (Table 1). Studies utilised L-Glu directly or as MSG with concentrations ranging from 1.6 µM to 100 mM. L-Glu was prepared by dissolving in culture media directly or as a stock solution in phosphate-buffered saline (PBS) or 1 M hydrogen chloride (HCl) or dimethyl sulfoxide (DMSO). Most of the treatments were for 24 h, although study treatments of 0.5 h to 6 d were also undertaken.

#### 3.1.1. L-Glu Exposure Reduces Neuronal Viability

L-Glu effects on neuronal cell proliferation, viability, and cytotoxicity utilised several assay types that measured cellular metabolic activity, cytolysis, DNA fragmentation, the release of structural proteins, and stress markers, along with cell death via apoptosis or necrosis. L-Glu administration to differentiated and undifferentiated SH-SY5Y cells resulted in decreased cell viability in a concentration-dependent manner (over the concentration range of 5, 10, 20, 40, and 80 mM), with undifferentiated cells more vulnerable to L-Glu exposure [54]. The lowest concentrations shown to induce neurotoxicity were 250 µM L-Glu for differentiated SH-SY5Y cells [60], whereas for undifferentiated SH-SY5Y cells, 12.5–100 mM L-Glu for 3 h still resulted in a significant concentration-dependent reduction of cell viability [56,62,66]. Similarly, an 8 h incubation of 15–25 mM L-Glu to undifferentiated SH-SY5Y cells caused a significant reduction in cell viability [57]. Reduced cell viability was similarly observed for undifferentiated SH-SY5Y cells after exposure to L-Glu at concentrations from 1 to 100 mM for 12 or 24 h [51,52,53,55,58,59,63,64,66]. However, by contrast, undifferentiated SH-SY5Y cells were not affected at concentrations lower than 40 mM L-Glu for 24 h, according to a study by de Oliveira et al. (2019) [61].

L-Glu induced a loss of cell viability that reflected the length of exposure time from 2 to 24 h [53,57], but toxicity was greater after 24 h rather than 48 or 72 h [65]. Human fetal neurons displayed progressive loss throughout 6 d after L-Glu, as evidenced by microscopic examination [73]. For other primary fetal cortical neurons and stem cells, a significant reduction of cell viability was only observed in older cultures [71,74], as well as in HCN-1A and IMR-32 cell lines for which a significant loss of viability was observed after a 24 h incubation [68,70].

L-Glu caused a loss of neuronal membrane integrity and release of cytosolic lactate dehydrogenase (LDH) into the cell culture medium as an alternative means to quantify cell viability. L-Glu at a concentration range of 0.06 to 10 mM significantly increased LDH release in primary or neuronal cell lines [51,59,64,67,68,70,73]. An L-Glu concentration range from 0.8 to 50 mM applied to human neural stem cells caused significant LDH leakage in a concentration-dependent way and was maximal at 12.5 mM, indicative of saturation of cytotoxicity [72].

Another marker of L-Glu toxicity was DNA fragmentation, and this increased after 80 mM L-Glu administration for 24 h to SH-SY5Y cells [61].

L-Glu exposure resulted in structural damage, such as reduced expression of neurofilament 200 (NF200) protein and a marker of plasticity, polysialylated neural cell adhesion molecule (PSA-NCAM) [68]. L-Glu also triggered the expression of endoplasmic reticulum (ER) stress markers and other stress response proteins. The expression level of ER stress-related proteins, such as CCAAT/enhancer-binding protein homologous protein (CHOP), glucose regulatory protein 78 (GRP78), and caspase-4, was significantly increased after 10 mM L-Glu addition to SH-SY5Y cells for 24 h [64]. Additionally, the stress signal 70 kDa heat shock protein (HSP70) was elevated following 0.25 and 0.5 mM L-Glu addition to IMR-32 cells [68].

L-Glu induced cell apoptosis at a concentration of 8 [51], 10 [59,64], 20 [54,58], 25 [57], 30 [56], 50, and 80 mM [55]. However, one study reported that no cellular apoptotic response was observed in response to 5- and 10-mM L-Glu treatment [74]. L-Glu-induced apoptosis was associated with an elevation of the expression of c-fos and c-jun genes in SH-SY5Y cells [67].

Exposure to L-Glu at 20 and 50 mM caused an increase in the percentage of necrotic neurons and upregulated the expression of the key signalling molecule, necrosis receptor-interacting protein (RIP) kinase 1, but not RIP kinase 3 [57].

#### 3.1.2. L-Glu Exposure Impairs Cellular Oxidant Defence and Stimulates Oxidative Stress

Five cell-based studies reported that L-Glu exposure impaired the endogenous antioxidant defence system. Relatively high concentrations of L-Glu (above 10 mM) caused significantly decreased activities of superoxide dismutase (SOD) and catalase (CAT), and depleted cellular glutathione (GSH) [59,62,64,65]. However, one study reported that although 15 mM L-Glu significantly reduced CAT activity, there was only a slight and nonsignificant decrease in SOD activity [66]. Research using a human neuron model reported that L-Glu exposure at 10, 20, and 30 mM for 3 h also significantly impaired the expression of the antioxidant defence Nrf2/HO-1 (nuclear factor erythroid 2-related factor-(Nrf2-)/heme oxygenase-1 (HO-1)) axis [56].

At pathological concentrations of L-Glu (≥10 mM), there was also an induction of cellular oxidative stress. Eight studies reported a significant rise in the production of ROS [54,55,56,58,59,61,63,64]. Hydrogen peroxide (H_2_O_2_) levels also rose after L-Glu application [66], and plasma levels of malondialdehyde (MDA), an indicator of lipid peroxidation (LPO), were raised after 10, 15, and 80 mM L-Glu treatments [59,61,64,65,66]. Incubation with 10 and 80 mM L-Glu induced protein oxidation, evidenced as increased levels of protein carbonyl content (PCC) [59,61]. Furthermore, L-Glu-induced protein oxidation was detected via depletion of protein thiols and increased protein nitration or 3-nitrotyrosine content [61]. DNA oxidative damage was also evident as increased 8-oxo-2′-deoxyguanosine (8-oxo-dG) content [61].

#### 3.1.3. L-Glu Enhances Acetylcholinesterase (AChE) Activity

A single study investigated the in vitro effects of L-Glu on AChE activity in differentiated SH-SY5Y cells, and this increased significantly after a 100 mM exposure for 3 h [62].

#### 3.1.4. L-Glu Exposure Triggers Mitochondria Dysfunction and Neuronal Apoptosis

L-Glu administration at concentrations of 8, 10, 20, 30, 80, and 100 mM resulted in mitochondria impairment and increased mitochondria-related apoptotic factors, such as cleaved poly (adenosine diphosphate (ADP)-ribose) polymerase (PARP) [61,62], caspase-3 [56,59,61,62,64,65], caspase-9 [58,61,65], and Bcl-2-associated X protein (Bax) [51,58,59,61,62]. L-Glu also triggered the downregulation of antiapoptotic B-cell lymphoma-2 (Bcl-2) expression [51,58,59,62], upregulation of the Bax/Bcl-2 ratio [64], and release of other proapoptotic proteins, such as cytochrome c [58,61].

Levels of phosphorylated mitogen-activated protein kinase (MAPK) forms: p38 [59,62,64], p54 (c-jun N-terminal kinase) (JNK) [59,64], and p42/44 extracellular signal-regulated kinases (ERKs) [59] were upregulated after L-Glu exposures, contributing to the activation of neuronal apoptosis.

The activity of the mitochondrial enzymes aconitase, α-ketoglutarate dehydrogenase (α-KGDH), succinate dehydrogenase (SDH), complex I (NADH dehydrogenase), and complex V was significantly reduced after exposure to L-Glu at 80 mM [61]. Adenosine triphosphate (ATP) levels declined after L-Glu exposure [59,61,64], as did the mitochondrial membrane potential (MMP) [57,58,59,61,64]. However, one study reported a heightened MMP at L-Glu concentrations of 25 or 50 mM, and this may contribute to neuronal cell death [57]. Mitochondrial protein carbonyl content and 5′ AMP-activated protein kinase (AMPK) activity were enhanced due to L-Glu exposure, indicative of oxidative stress and apoptosis, respectively [56,59].

#### 3.1.5. L-Glu Exposure Stimulates Excitotoxicity and Alters Neuronal Calcium Levels

Six in vitro studies considered the role of L-Glu in excitotoxicity via its action as an excitatory neurotransmitter to damage neurons through overactivation of its receptors. After 24 h exposure to 1 and 0.1 mM L-Glu, there was an excessively high intracellular accumulation of Ca^2+^ [67,70]. Similarly, there was a concentration-dependent elevation of intracellular Ca^2+^ concentration after exposure to L-Glu (15–25 mM) for 1 h, but this was not statistically significant at concentrations of 10 and 50 mM [57]. Ca^2+^ ion influx into neurons was significantly increased at 10 mM L-Glu [64]. Likewise, 6- and 8-week-old cultures of human embryonic stem cell (HESC)-derived neurons developed increasing Ca^2+^ influx in response to extracellular L-Glu application, in contrast to early week neurons, which were unresponsive to L-Glu [71]. These 8-week cultures had increased expression of NMDA and AMPA receptor subunits [71]. A functional assessment of GABAergic neurons displayed concentration-dependent decreases in ^3^H-GABA uptake in response to exposure to L-Glu (1.6 to 5000 µM) for 6 d [73].

#### 3.1.6. L-Glu Exposure Triggers Neuroinflammation

Five cell-based studies suggested an association of L-Glu with neuroinflammation. There was increased expression and activity of early matrix metalloproteinase (MMP) following L-Glu [68]. Activation of proinflammatory markers, such as nuclear factor kappa-light-chain-enhancer of activated B cells (NF-kB) and cyclooxygenase-2 (COX-2), was upregulated after 30 and 2.5 mM L-Glu exposures, respectively [56,72]. In addition, inducible nitric oxide synthase (iNOS) was increased following exposure to 2.5 mM L-Glu [72]. There was a marked increase in Nod-like receptor protein 3 (NLRP3) protein inflammasomes in response to L-Glu stimulation [64]. At 2.5 and 10 mM L-Glu, interleukin 6 (IL-6) was induced [64,72]. Tumour necrosis factor-α (TNF-α) was increased at L-Glu concentrations of 2.5 and 80 mM [65,72] and interleukin 1β (IL-1β) at 10 mM L-Glu [64]. Other cytokine mediators of inflammation, such as transforming growth factor-β (TGF-β) and high-mobility group box 1 (HMGB1), were significantly increased in response to L-Glu [72].

#### 3.1.7. L-Glu Exposure Affects the Morphological Characteristics of Neurons

Nine studies reported morphological alterations to neurons following L-Glu administration, with alterations to nuclear size, chromatin condensation, and nuclear fragmentation [51,53,55,62,66,74], as well as signs of toxicity, particularly as cell body shrinkage and dendritic retraction [51,53,54,55,68,70].

#### 3.1.8. Protein Aggregation

One article that investigated the relation between L-Glu treatment and toxic protein aggregation reported that SH-SY5Y exposed to 1 mM for 6 h resulted in a build-up of tau protein phosphorylation, but this was not statistically significant [52].

### 3.2. In Vivo Studies Evaluating L-Glu Toxicity

In vivo (animal) data focused on the direct impact of L-Glu on neuronal molecular processes that resulted in disease, such NDDs. Different animal models were used to evaluate L-Glu neurotoxicity, including albino Wistar rats (n = 22) [27,33,36,75,76,77,78,79,80,81,82,83,84,85,86,87,88,89,90,91,92,93], Sprague–Dawley rats (n = 13) [56,94,95,96,97,98,99,100,101,102,103,104,105], other rats (n = 1) [106], Swiss albino mice (n = 3) [28,93,107], wild-type mice (n = 5) [108,109,110,111,112], Kunming mice (n = 2) [113,114], CD-1 mice (n = 1) [115], *Caenorhabditis elegans* nematodes (n = 1) [116], and ephyrae of *Aurelia aurita* (n = 1) [117]. Moreover, there were different routes of L-Glu administration in these in vivo studies: orally (*p.o.*) (n = 10) [28,33,83,87,89,90,93,106,107,115], subcutaneous (*s.c.*) injection (n = 12) [33,36,75,78,79,81,86,88,93,96,102,105], intraperitoneal (*i.p.*) injection (n = 8) [27,56,77,84,85,91,92,103], maternal intragastric (*i.g.*) (n = 2) [113,114], stereotactic injection via intrastriatal or cerebral cortex injection (n = 12) [80,82,94,97,98,99,101,104,108,109,110,112], microdialysis (n = 3) [76,95,100], brain infusion via a cannula (n = 1) [111], and also L-Glu to animals via living media (n = 2) [116,117]. Furthermore, the data from current research found that different parts of the nervous system were examined to evaluate L-Glu neurotoxicity. The number of studies used the whole of brain homogenates [27,84,88,91,92,96,103,115], cerebral cortex [33,36,78,81,83,85,89,95,97,98,99,101,104,106,107,108,112], hippocampus [27,28,33,56,79,84,85,86,87,92,102,113,114], striatum [36,76,80,82,94,100,108,109,110], cerebellum [77,85], forebrains [90], hypothalamus [75,93], circumventricular organs [75], spinal cord [105,111], cerebral hemisphere, brain stem, diencephalon [77], pituitary gland [96], neural circuits [116], and rhopalia [117]. Table 2 shows the results of in vivo research classified based on the animal’s species and strain.

#### 3.2.1. Administration of L-Glu Directly to Animals

##### L-Glu Administration Reduces Neuronal Viability

Neuron injury enzyme markers, such as serum creatine phosphokinase (CPK) and creatine phosphokinase isoenzymes BB (CPK-BB), were significantly higher in the brains of rats injected *s.c.* with L-Glu [88]. LDH released from damaged tissue was significantly elevated in the brains of rats injected with L-Glu *i.p.* or *s.c.* [88,91]. Additionally, the Ki-67 protein, a marker of actively proliferating neurons, was significantly declined in brain tissue after L-Glu administration to rats *i.p.* [92]. High fluoro jade B (FJB)-stained neurons and overactivation of poly (adenosine diphosphate (ADP)-ribose) polymerase (PARP-1) were observed, two labels of degenerating neurons, as a result of a high dose of exogenous L-Glu to the brain [102]. There was also a significant reduction in brain-derived neurotrophic factor (BDNF) following L-Glu treatment at 17.5 mg/kg dose *p.o.*, although this was not significant at a 6 mg/kg dose [106].

##### L-Glu Administration Impairs Cellular Oxidant Defence and Stimulates Oxidative Stress

Delivery of a high dose (≥17.5 mg/kg) of L-Glu by either *p.o.* or *i.p.* routes significantly reduced brain SOD and CAT levels [28,77,84,85,89,106,107], although a relatively low L-Glu dose (6 mg/kg) was without effect on these enzymes [106]. However, low-dose L-Glu (*s.c.* injection of 5 mg/kg) resulted in a significantly increased expression of SOD and CAT genes [88]. L-Glu at 4 g/kg by *s.c.* and *i.p.* routes significantly reduced the activity of SOD in the CNS [92,105]. Conversely, a relatively high dose of L-Glu (2 g/kg *i.p.*) was without effect on SOD activity [103]. However, a high dose of L-Glu (4 g/kg injected *s.c.*) induced CAT [75], whereas 4 g/kg via an *i.p.* injection reduced CAT activity [27].

Brain GSH level was significantly declined after L-Glu treatment by *p.o.* [89,106,107], *s.c.* [88], or *i.p.* routes [27,56,77,84,85]. In addition, a relatively high oral dose of L-Glu (2 g/kg) was also able to reduce cellular GSH levels but not significantly [115].

A similar contrary result was observed after L-Glu was taken orally (17.5 mg/kg), as this induced inhibition of glutathione peroxidase (GPx) activity [106], while a high dose (4 g/kg) administered *i.p.* resulted in a significant boost of GPx activity [77]. Similarly, glutathione-S-transferase (GST) activity was inhibited after L-Glu treatment *i.p.* [27]; however, its activity and gene expression were significantly increased in response to *s.c.* L-Glu injection [88].

The modest antioxidant uric acid was significantly reduced but only after a significant drop in the concentration of other antioxidants [77]. This study also showed a significant elevation in brain uric acid content in response to *i.p.* L-Glu injection [77].

L-Glu *i.p.* resulted in the suppression of the transcription factor nuclear factor E2-related factor 2 (Nrf2), which regulates the expression of antioxidant proteins [56]. There was also a significant decline in thiol levels in *p.o.* and *i.p.* L-Glu-treated groups [27,106], but not after a low *p.o.* dose (6 mg/kg) [106]. However, the activities of the brain antioxidant enzymes myeloperoxidase (MPO) and xanthine oxidase (XO) were unaffected following L-Glu administration [106].

Administration of L-Glu in vivo by *p.o.*, *s.c.*, or *i.p.* routes triggered the accumulation of LPO products [27,75,77,84,85,88,89,92,105,107,115], but L-Glu *s.c.* at 0.004 mg/kg dose showed no significant differences in LPO levels within the frontal cortex [75]. Furthermore, a single study reported significantly reduced LPO levels following L-Glu injection *i.p.* [103].

There was a significant elevation of reactive nitrogen species (RNS) after L-Glu via *p.o.* or oral cannula or *i.p.* injection [27,28,89]. Relatively high levels of ROS were apparent after L-Glu was given *i.p.* [56]. L-Glu administrated *p.o.* also significantly elevated 8-hydroxy-2′-deoxyguanosine (8-OHdG) levels in rat brains, a biomarker of oxidative damage to DNA [89].

##### L-Glu Administration Influences Acetylcholinesterase (AChE) Activity

AChE activity was significantly reduced at relatively higher doses of L-Glu (17.5 mg/kg, *p.o.* [106] or 100 mg/kg, *p.o.*) [89], whereas a low dose of L-Glu (6 mg/kg, *p.o.*) did not affect AChE activity [106]. However, by contrast, AChE activity was significantly enhanced by L-Glu at 4 mg/kg (*p.o.*) or 5 mg/kg (*s.c.*) [87,88].

##### L-Glu Administration Influences Neurotransmitter Levels

L-Glu administration (4 and 100 mg/kg, *p.o.*; 10 mg/kg, *i.p.*; 5 and 10 mg/kg, *s.c.*) significantly elevated brain L-Glu neurotransmitter levels [56,87,89,102], but not at doses of 10, 20, 40, and 80 mg/kg, *p.o.* [28], or 2 g/kg, *p.o.*, or 4 g/kg, *s.c.* [33]. Dopamine levels were significantly reduced after a 17.5 mg/kg dose of L-Glu (*p.o.*) and 100 mg/kg dose via oral gavage [89,106]. Similarly, brain serotonin was significantly elevated after L-Glu at 100 mg/kg, *p.o.* [89], but also significantly declined following L-Glu at 17.5 mg/kg, 0.83 g/kg, and 1.66 g/kg, *p.o.* [90,106]. The neurotransmitters noradrenaline and adrenaline also declined after administration of L-Glu at 17.5 mg/kg (*p.o.*) [106]. Catecholamine (noradrenaline and adrenaline), dopamine, and serotonin levels were not altered at an L-Glu dose of 6 mg/kg *p.o.* [106].

##### L-Glu Administration Triggers Neuronal Apoptosis

The administration of L-Glu by *s.c.* and *i.p.* routes induced activation and phosphorylation of AMP-activated protein kinase (AMPK) in some [56,102] but not all studies [33]. L-Glu (*s.c.*) induced a significant upregulation of phosphorylated activating transcription factor 2 (ATF2) expression and associated p38 MAPK activity [78,86,105]. The number of terminal deoxynucleotidyl transferase (dUTP) nick end labeling (TUNEL)-positive neurons (indicative of apoptosis) was significantly increased after L-Glu treatment (*s.c.* or *i.p.*) [36,56,79,86]. There was also a significant upregulation of the proapoptotic Bax and downregulation of antiapoptotic Bcl-2 genes in rats treated with L-Glu by *s.c.* [88,102] and *i.g.* [113] routes. However, one study reported that Bcl-2 mRNA expression in the brain was increased as a consequence of *s.c.* L-Glu injection [81].

Apoptosis signalling caspase-3 protein was significantly increased in its expression in the brain in response to L-Glu injected *s.c.* [102,105], *i.p.* [56,91,92] in rat, or *i.g* in mice. [113]. L-Glu injected *i.p.* or *s.c.* initiated a significant rise in the cytochrome c gene expression [91] and release [102] in rat brain tissue. Administration of L-Glu *p.o.* and *s.c.* induced a rise of the Fas ligand as an apoptosis mediator [33,81].

##### Excitotoxicity, Calcium Level, and Other Ions in the Brain

Rats administered L-Glu via *p.o.* and *s.c.* routes exhibited significantly increased expression of specific NMDAR subunits, including NMDA2B and NR1, in several brain areas [78,86,87]. In addition, *s.c.* and *i.p.* L-Glu induced significantly higher expression of AMPA receptor subunits, GluR1 [86] and GluR2 [56,78,81]. Furthermore, *s.c.* L-Glu treatment significantly increased the level of expression of the GluR2 transcription regulator known as neuron-restrictive silencer factor (NRSF) mRNA levels or RE1-silencing transcription factor (REST) [81,86]. There was also a nonsignificant increase in the expression of the metabotropic glutamate receptor 5 (mGluR5) gene in the brain in response to *p.o.* L-Glu [87]. As a consequence of L-Glu excitotoxicity, its administration *i.p.* into animals resulted in a significant reduction in the level of the inhibitory neurotransmitter gamma-aminobutyric acid (GABA) [84].

Significant elevation of Ca^2+^ levels was observed in the brains of rodents treated with L-Glu *p.o.* [89] or *i.p.* [56,84,92]. In the L-Glu *i.p.* treated group, the Ca^2+^ level was detected by strong calretinin immune reactivity in neurons and upregulation of the expression level of Ca^2+^/calmodulin-dependent protein kinase II (CaMKII) [56,92]. This was associated with heightened Na^+^ levels and reduced K^+^ levels [84,88,89].

##### Neuroinflammation

The brain or spinal cord tissue of rats exhibited a significant elevation in the levels of neuroinflammatory cytokines, including TNFα, after *p.o.*, *s.c.*, or *i.p.* applied L-Glu [36,56,79,87,105]. Additionally, relatively elevated levels of the cytokines IL-1ß and IL-6 were detected in the cervical spinal cord and brain of rats *s.c.* injected with L-Glu [36,79,105]. Likewise, *s.c.* L-Glu-treated rats displayed a substantial rise in the amount of spinal cord interferon-γ (IFN-ɣ) with a significant decline in the anti-inflammatory cytokine IL-10 as compared with controls [105].

Administration of L-Glu *i.p.* to rats revealed a significant induction of glial cell activation, which was identified by an upsurge in the glial fibrillary acidic protein (GFAP) immunodetection [56,92] and a marker for microglial activation, Iba-1 [56]. A study that analysed the morphometrics of GFAP-stained astrocytes showed that *s.c.* application of L-Glu resulted in considerable shrinkage of the astrocyte surface area in the spinal cords of rats [105]. Moreover, gliosis was detected in response to *i.p.* L-Glu administration to rats, the glial cells’ reactive change in response to brain damage [92]. L-Glu via *i.p.* and *p.o.* routes at doses of 10 and 17.5 mg/kg induced the activities of the proinflammatory mediators COX-2 [56,106] and prostaglandin E2 (PGE2), respectively [106]. However, no difference was found in these proinflammatory mediators after a low *p.o.* L-Glu dose of 6 mg/kg [106].

In addition, the activity and phosphorylation of proinflammatory nuclear factor kappa-light-chain-enhancer of activated B cells (NF-κB) were upregulated in response to *i.p.* L-Glu rat injection [56]. Furthermore, L-Glu given by oral gavage to animals was associated with inflammation in mouse brains via the enhanced activity of phospholipase A2 (PLA2) [115].

##### Histology Alteration

Histological and histomorphometric changes in several brain areas were the result of *p.o.*, *s.c.*, or *i.p.* L-Glu administration, consistent with neuronal degeneration [96] via necrosis or apoptosis and nuclear pyknosis [28,78,83,84,86,89,92,93,105,106,107]. In addition, the brain tissue following the application of L-Glu *p.o.*, *s.c.*, or *i.p* exhibited areas of haemorrhage [106], congestion [89,106], accumulation of inflammatory cells [78,83,92,96], oedema [84,89], neuronal cytoplasmic vacuolisation [28,78,92,105,107], and neuronal eosinophilia [84]. Additionally, *i.p.* L-Glu treatment caused morphological changes to neuropil fibres [27,85] and neuronal karyorrhexis [84]. Treatment with L-Glu by *i.g.* route caused brain tissue alterations, including oedema, neuron death via necrosis, and hyperplasia [113,114]. However, two studies indicated that L-Glu *p.o.* (10 and 20 mg/kg) and *i.p.* (2g/kg) had no effects on the neuronal features of brain tissue [28,103].

##### Behaviour and Cognitive Function

The negative impact of *p.o.*, *s.c.*, or *i.g.* L-Glu administration on cognitive functions and the induction of the impairment of learning ability has been extensively demonstrated [33,90,91,113,114]. Similarly, poor memory retention was observed in animals following L-Glu, *i.p* [85,92], causing locomotor impairment (L-Glu, *i.p*. or *i.g.*) [27,84,92,113] or coordination abnormalities (L-Glu, *i.p.*) as well [85]. In contrast, one study reported that treatment by L-Glu *p.o.* or *s.c.* caused neurobehavioral deficits but without any sign of motor or coordination abnormalities [33]. In addition, L-Glu, *p.o.* or *s.c.*, triggered behavioural phenotypic changes, including increased anxiety [33,107], while aggressiveness and loss of muscle strength were detected after L-Glu, *i.p.*, treatment [84,85].

##### Protein Aggregation

A significant increase in β-amyloid (Aβ) protein accumulation by more than twofold in rat brain tissue was reported after L-Glu administration through *i.p.* (7 days) or *p.o.* (2 months) or *p.o. and s.c.* (10 days) routes [33,89,91].

##### Brain Weight

One study indicated that relative brain weight was increased significantly after *p.o.* L-Glu was introduced at 40 and 80 mg/kg in mice [28]. However, a higher *p.o.* dose of 1000 mg/kg L-Glu caused a significant decrease in relative brain weight in mice [107]. This was supported by another study showing that the total protein of brain tissue was reduced significantly in high-dose 2 g/kg, *i.p.*, L-Glu-treated rats [85].

#### 3.2.2. L-Glu Administration Directly to Animal Brains

A total of 16 in vivo studies administrated L-Glu directly into brain tissue through microdialysis, stereotactic injection, or brain infusion canal (Table 2).

##### Antioxidant and Oxidative Stress Markers

Four studies reported that L-Glu stereotactic administration into the cerebral cortex induced significant reductions in the endogenous antioxidant defences: SOD, CAT, GSH, and glutathione reductase (GR) [97,99,101,104]. Similarly, cerebral cortex GSH levels considerably declined after L-Glu stereotactic injection [98]. In addition, the cerebral cortex GPx was reduced in L-Glu-treated animals [101]. Moreover, the expression of the antioxidant regulatory proteins nuclear factor erythroid 2-related factor 2 (Nrf2), glutamate cysteine ligase catalytic subunit (GCLC), and heme oxygenase-1 (HO-1) was significantly decreased in the cerebral cortex of the L-Glu-treated group [104].

L-Glu administration through microdialysis (15 mM) or stereotactic injection (1 µL of 1 µM or 1 M) significantly enhanced the LPO level in the cortex and striatum of rats [80,95,97,99,101,104]. Similarly, A 1.5 mM L-Glu solution increased the LPO level by approximately 100%, but this was not statistically significant [95]. There is also experimental evidence of elevated LPO levels detected 3 h after intrastriatal injection with L-Glu [80]. Furthermore, stereotactically injected L-Glu induced significantly high levels of ROS formation in the brain striatum and cerebral cortex [97,98,99,101,104,110]. There was also elevated nitric oxide (NO^•^) production in the cerebral cortex after stereotactic L-Glu administration [97,99,101,104] and increased mRNA expression of iNOS [98,99,104] and nNOS [98,99,101,104]. L-Glu exposure significantly enhanced NADPH oxidase (NOX) activity in the cerebral cortex and striatum [101,104,110]. An elevated number of neurons positive for the marker of oxidative damage, nitrosylated proteins, were detected in the striatum of animals subjected to L-Glu [110]. Cortical neuron peroxynitrite (ONOO^−^) production level was significantly higher in animals injected with L-Glu [98]. The perfusion of L-Glu into the striatum induced the formation of 2,3-dihydroxybenzoic acid (2,3-DHBA), reflecting a significant increase in the levels of hydroxyl radicals (HO^•^) [76].

##### Neurotransmitter Levels

Introduction of L-Glu into the striatum via microdialysis induced intracellular L-Glu build-up in astroglia and alanine but a reduced glutamine level in the neurons [100].

##### Mitochondrial Dysfunction and Apoptosis

L-Glu, stereotactically injected into the cerebral cortex, significantly reduced the mitochondrial membrane potential of cortical neurons [97,98,99,101]. In addition, a significantly diminished mitochondrial ATP level was recorded in animals receiving L-Glu in one study [104], but this was not statistically significant in another study [80]. L-Glu perfusion into the left lateral ventricle significantly caused mitofusin 2 (MFN2) decline and mitochondrial fragmentation in spinal cord neurons [111]. Furthermore, L-Glu stereotactically injected into the cerebral cortex induced mitochondrial dysfunction by causing sodium potassium adenosine triphosphatase (Na^+^-K^+^-ATPase) level reduction and a decline in the activity of mitochondrial cytochrome c oxidase [104].

L-Glu administration via stereotactic injection or infusion cannula induced the proapoptotic protein caspase-3 levels in the brain and spinal cord neurons [98,99,101,104,111]. In contrast, one study reported that caspase activation was not affected following intrastriatal L-Glu injection [82]. Animals stereotactically injected with L-Glu exhibited higher production of the proapoptotic protein caspase-9 [98,99,104]. Stereotactically injected L-Glu reduced antiapoptotic Bcl-2 gene expression and elevated pro-apoptotic Bax gene expression, such that the Bcl-2/Bax ratio was considerably reduced [98,104]. Additionally, the levels of phospho-extracellular signal-regulated kinase (pERK) rose, leading to apoptosis in the cerebral cortex after stereotactic administration of L-Glu [101].

##### Calcium Level

Stereotactic administration of L-Glu into the cerebral cortex resulted in significant elevation of intracellular levels of Ca^2+^ [97,98,99,101], and the Ca^2+-^dependent protease calpain was also triggered in response to L-Glu stereotaxic injection [82,104,110].

##### Neuroinflammation

Studies into the influence of L-Glu on neuroinflammation observed that its introduction into the brain triggered reactive astrogliosis [100] and microglial activation [110]. In contrast, a report evaluating continuous L-Glu administration concluded that it did not activate astrocytes or microglia in spinal cords [111]. However, stereotactic injection of L-Glu induced the production of the proinflammatory cytokines TNF-α, IL-1β, and IFN-ɣ in the cerebral cortex [97,99,101,104].

##### Histological Abnormalities

Studies that examined brain sections showed lesions in the striatum after L-Glu was introduced at concentrations of 500 nM and 0.3 M [82,94]. Lesions were observed after histological examination of sections from the brain cortex and striatum following L-Glu exposure at concentrations of 500 nM, 1 µM, and 0.25 M [80,108,109,110,112]. Data showed a higher number of degenerating fluoro jade-positive neurons in the striatum of L-Glu-exposed rodents [108,110]. A histological study indicated that the cerebral cortex pyramidal neurons were significantly smaller in size in L-Glu-treated rats [97,98]. Another study reported an elevated number of neurons with positive terminal deoxynucleotidyl transferase (dUTP) nick end labeling (TUNEL) in the cerebral cortex, indicating DNA fragmentation [104]. However, brain infusion with L-Glu at 10 mM did not trigger any pathohistological features or alteration in brain tissue [111].

##### Behaviour and Cognitive Function

Stereotactic injection of 0.25 M L-Glu into the parietal cortex had no effects on learning ability when assessed via a Morris water maze (MWM) test [112]. Similar results were found in a study reporting that a perfusion of 50 mM L-Glu via microdialysis into the striatum did not alter the locomotor activity or circling behaviour [76].

##### Electroencephalogram (EEG)

Research assessing the electrical activity of the brain reported that after 500 nM/0.5 μL L-Glu stereotaxic injection, mice brains exhibited regular electrical activity [108].

##### L-Glu Administration Directly to an Animal’s Living Media

L-Glu addition to the media of living animals induced oxidative stress through increased production of ROS [116,117], superoxide (O_2_^−^**^•^**) [116], and NO^•^ [117]. Depletion of GSH was detected following L-Glu administration to the growth medium of the *Caenorhabditis elegans* nematode [116]. Furthermore, 5 mM L-Glu in the artificial seawater of the ephyrae of *Aurelia aurita* raised the Ca^2+^ level in the animal’s sensory organ, rhopalium [117]. Administration of L-Glu into the media of *C. elegans* and the ephyrae of *Aurelia aurita* also affected the locomotory ability of these animals [116,117].

## 4. Discussion

Several organisations, such as the USA Food and Drug Administration (FDA), the European Food Safety Authority (EFSA), and the Joint FAO/WHO Expert Committee on Food Additives (JECFA), have continually reasserted the safety of L-Glu [47]. However, the EFSA panel re-evaluated L-Glu safety in 2017 and suggested that exposure to L-Glu that exceeded the acceptable daily intake (ADI) of 30 mg/kg bw per day in all age groups is linked to adverse health effects [44,118]. Nutritional analytical studies suggest that the daily intake of free L-Glu in humans is greater than 1 g and can reach 10 g/day, equivalent to 170 mg/kg bw, for a 60 kg person [119,120]. As a result of the widespread global consumption of L-Glu, it remains a contentious issue, as excessive exposure has been associated with defects in neurophysiological function in both human and animal research [35].

This review considered whether the levels of L-Glu have pathogenic consequences for neurons and could therefore contribute to the development of certain NDDs, such as AD, PD, MS, ALS, and HD. We have highlighted the experimental evidence that supports the involvement of L-Glu in a number of toxic cellular mechanisms. L-Glu has an impact upon neuronal viability, oxidative stress, endogenous antioxidant defence, neuroinflammation, neurotransmitter levels, aberrant protein accumulations, excitotoxicity, mitochondrial dysfunction, intracellular Ca^2+^ levels, neuronal morphology, animal behaviour, and cognitive function.

However, two major mechanisms underpin L-Glu toxicity: receptor-mediated excitotoxicity and non-receptor-mediated oxidative stress, and they are integrated in parallel in neurons, as shown in Figure 2 [121,122]. In the receptor-mediated excitotoxicity, there is an excess of Ca^2+^ influx into neurons as a result of L-Glu overactivation of its receptors (AMPAR, NMDAR, and kainic acid receptors (KARs)) [32,121] and due to the activation of voltage-dependent Ca^2+^ channels (VDCCs) [123]. Furthermore, stimulation of mGlu receptors increases the synthesis of inositol triphosphate (IP_3_) and the release of Ca^2+^ from endoplasmic reticulum (ER) stores [121]. Pathologically high levels of Ca^2+^ ions result in the activation of Ca^2+^-dependent protease enzymes, which can degrade proteins in neurons, such as cytoskeletal proteins, and generate oxidative stress [121]. Oxidative stress promotes neuron death by damaging crucial cell components, such as the cell membrane, proteins, and DNA [121]. Ca^2+^ also mediates mitochondrial dysfunction, resulting in the release of proapoptotic proteins [121], and causes ER stress-induced cell death [124]. The cystine/glutamate antiporter system (Xc) can be involved in the non-receptor-mediated oxidative stress [122]. An excess of extracellular L-Glu blocks the glutamate/cystine antiporter system Xc-, and this results in reduced cysteine, which is a required component for the production of the major cellular antioxidant, glutathione (GSH) [122]. This causes oxidative glutamate toxicity, also known as oxytosis, which causes cell death by oxidative stress via reactive oxygen species (ROS) accumulation [122]. Oxidative stress promotes the activation of numerous pathways, resulting in proinflammatory cytokine production and neuroinflammation (Figure 2) [125].

### 4.1. Common L-Glu Neurotoxic Pathways In Vitro and In Vivo

#### 4.1.1. Cellular and Molecular Changes

The experimental evaluation of both cell-based studies and animal (in vivo) findings supports a toxic effect of L-Glu with diminished neuronal viability [51,52,53,54,55,56,57,58,59,60,62,63,64,65,66,68,70,71,73,74,88,92,102,106] and release of LDH [51,59,64,67,68,70,72,73,88,91]. L-Glu induces apoptosis via influencing the activity of several apoptotic factors, such as caspases-3 and 9, cytochrome c, and activation of intracellular signalling pathways, including MAPKs, AMPK, PARP, Fas ligand, Bax, and Bcl-2 [33,51,58,59,62,81,98,104,113] (Table 2). Furthermore, cell growth and proliferation were perturbed, and markers of cellular structural changes and injury were observed [64,68,88,92,106].

Cellular damage from L-Glu was in part derived from the generation of redox stress and depletion of the antioxidant defence system. This included reduced SOD [28,59,62,64,65,77,84,85,89,97,99,101,104,106,107], CAT [28,59,62,64,65,77,84,85,89,97,99,101,104,106,107], and GSH activities [27,56,59,62,64,65,77,84,85,88,89,97,98,99,101,104,106,107,116]; inhibited GR [97,99,101,104], GPx [101,106], and GST [27]; impact upon the expression of antioxidant regulatory proteins, Nrf2 [56,104], HO-1 [56,104]; and GCLC [104], and thiol levels [27,106].

There were significant accumulations of ROS [54,55,56,58,59,61,63,64,97,98,99,101,104,110,116,117], RNS [27,28,72,89,97,98,99,101,104,110,117], O^2−•^ [116], HO^•^ [76], and H_2_O_2_ content [66], as well as LPO build-up, following L-Glu exposure [27,59,61,64,65,66,75,77,80,84,85,88,89,92,95,97,99,101,104,105,107,115]. Moreover, oxidative damage to proteins [59,61,110] and DNA [61,89] was induced in vitro and in vivo after L-Glu.

L-Glu induced a significant elevation in AChE activity in vitro and with relatively high L-Glu dosing in vivo [62,87,88]. However, in contrast, AChE activity was reduced in other in vivo studies conducted at lower L-Glu dosing [89,106]. A number of brain neurotransmitter levels were impaired after L-Glu application in vivo, such as L-Glu [56,87,89,100,102], dopamine [89,106], serotonin [89,90,106], noradrenaline, and adrenaline [106].

Both in vivo and in vitro studies demonstrated L-Glu impairment of mitochondrial function, which was observed through ATP [59,61,64,104] and MMP [57,58,59,61,64,97,98,99,101] diminution (Table 1 and Table 2). Furthermore, there was a reduction in the activity of a number of critical mitochondrial enzymes after L-Glu treatment [61,104]. L-Glu impaired mitochondrial fusion and triggered mitochondrial fragmentation in spinal cord neurons [111]. As a consequence of L-Glu excitotoxicity, its administration in vivo and in vitro triggered increased intracellular Ca^2+^ levels [56,57,64,67,70,71,84,89,92,97,98,99,101]. This resulted in Ca^2+^-dependent protease calpain production [82,104,110], along with increased expression of L-Glu receptor subunits, including NMDAR [71,78,86,87] and AMPAR [56,71,78,81,86] subunits. Although the change in mGluR5 expression was nonsignificant, upregulation of the L-Glu receptors’ expression resulted in Ca^2+^ release from intracellular stores, which triggered enzymatic overactivation, such as protein kinases, and led to cellular protein and membrane breakdown [87]. Furthermore, inhibitory GABA homeostasis was impaired after L-Glu exposure in vivo and in vitro [73,84].

Advanced activation of several proinflammatory markers, such as nuclear factor NF-kB [56], COX-2 [56,72,106], IL-6 [36,64,72,79,105], TNF-α [36,56,65,72,79,87,97,99,101,104,105], and IL-1β [36,64,79,104,105], were reported after L-Glu administration in vivo and in vitro. Additional inflammatory mediators, such as activation of MMPs [68], NLRP3 inflammasomes [64], TGF-β, and HMGB1, were a consequence of neuronal culture exposure to L-Glu [72]. Administration of L-Glu triggered the activity of the inflammatory biomarkers PGE2 [106], PLA2s [115], and IFN-ɣ [97,99,101,104,105], and a decline in the anti-inflammatory cytokine IL-10 was observed in L-Glu-treated rats [105].

Collectively, NDDs are characterised histopathologically by the accumulation of extracellular, cytosolic, or nuclear protein oligomers and fibrils, and the formation of these is influenced by an array of protein post-translational modifications (PTMs) [126]. For AD, the accumulation of extracellular Aβ peptide and intracellular hyperphosphorylated tau is thought to be toxic and contribute to neurodegeneration. L-Glu application triggered a significant increase in Aβ (1-42) accumulation in the brain tissue of rats [33,89,91], and increased levels of Aβ (1-40 and 1-42) were also observed in 3-month-old rats after neonate L-Glu administration [127]. L-Glu induced a nonsignificant increase in tau phosphorylation in vitro [52], but neonatal exposure to MSG increased tau phosphorylation in approximately 3-month-old rats [128] and 3- and 6-month-old mice [34,129]. L-Glu also stimulated increased tau translation [130].

Aberrant and neurotoxic protein aggregation could also be increased in response to other molecular mechanisms induced by L-Glu, including protein damage by redox stress such as oxidation of thiol groups and amino acids, and increased PCC [131], increased protein nitration, and altered levels of phosphorylation (Figure 3).

The common L-Glu neurotoxicity pathways reported in vitro and in vivo are summarised in Figure 4.

#### 4.1.2. Neural Structure Changes

Significant alteration in morphological characteristics was observed after L-Glu introduction in vivo and in vitro, including changes to the neuronal cell body, nucleus, and processes [28,51,53,54,55,62,66,68,70,74,78,83,84,86,89,92,93,97,98,104,105,106,107].

### 4.2. Other Neuropathology Observed after L-Glu Administration In Vivo

In addition to the common neuropathological outcomes evidenced with in vitro models, there were additional results obtained from in vivo models.

#### 4.2.1. Brain Structural Changes

Brain haemorrhaging and neuroinflammatory cell aggregation were observed in several in vivo studies following L-Glu induction [78,83,92,96], and L-Glu administration triggered gliosis in vivo [56,92,100,110].

#### 4.2.2. Changes in Behaviour and Cognition

Reports addressing the influence of excessive levels of L-Glu on animal behaviour and cognitive ability demonstrated its influence on memory [85,92], locomotor coordination [27,84,92,113,116,117], and learning ability impairment [33,90,91,113,114], as well as behavioural phenotype changes, including anxiety [33,107] and aggressiveness [84,85] (Table 2).

#### 4.2.3. Brain Weight and Homeostasis Changes

L-Glu systematic application caused a significant increase [28] or decrease [107] in the relative weight of mice brains. Alterations in brain ionic homeostasis, such as Na^+^ and K^+^ ion levels, were also observed [84,88,89].

### 4.3. Study Limitations

A common limitation in most of the models used to study L-Glu neurotoxicity is that the time used to monitor L-Glu neurodegeneration is typically short and included studies lasting hours and days to a few months, whereas in humans, excessive exogenous L-Glu exposure may be prolonged for years, in keeping with the gradual neuronal degradation that typifies NDDs. However, an understanding of acute L-Glu neurotoxicity may still be relevant to provide an insight into the molecular mechanisms that drive neuronal loss in NDDs and have an application to other diseases affected by L-Glu levels, such as stroke.

As summarised in Table 1, L-Glu neurotoxicity has been confirmed by human neuron studies in vitro. However, evidence from these studies showed that different neuronal cell lines have different reactions to L-Glu exposure. The SH-SY5Y cell line was the most resistant to L-Glu toxicity, whereas the most sensitive were primary neurons from human foetuses. This may be explained by the fact that in some cell lines, the L-Glu toxicity may occur through overactivation of specific L-Glu receptors, whereas in other cell lines lacking such receptors, the neurotoxicity may be manifested via alternative pathomechanisms, such as induction of redox stress [132]. Furthermore, the length of time associated with stimulation and the L-Glu concentration are important factors that will influence cell survival and death [132], and these vary between studies and, for some, may represent the application of supraphysiological L-Glu concentrations [133]. Additional variability between studies arose from differences in the preparation of L-Glu stock solutions either in cell culture media, PBS, or DMSO. Furthermore, in vitro studies invariably used isolated cells and therefore lack a blood–brain barrier (BBB) and the neuronal heterogeneity associated with the whole brain [134].

The findings of laboratory trials on animals reported many adverse impacts of L-Glu on neurological systems (Table 2). Most of the in vivo studies only considered acute neuronal damage, a few hours or days after the last treatment, whereas long-term effects and damage were much less often studied, with only five investigations performed that considered long-lasting neuronal damage after weeks to a month from the last L-Glu dose [33,75,77,92,96,112,113]. Surprisingly, neuronal damage was still evident after these potential periods of recovery, and the mechanisms of neurotoxicity were similar and sustained.

There were variability and limitations associated with the experimental design of some of the in vivo studies, including small sample sizes, broad and sometimes nonphysiological dosing, and a range of administration routes [47]. Furthermore, the current review has highlighted that the majority of in vivo studies have utilised male animals, and a range of parts of the nervous system have been investigated (cerebral cortex, hippocampus, cerebellum, cerebral hemisphere, brain stem, diencephalon, forebrains, hypothalamus, circumventricular organs, spinal cord, pituitary gland, neural circuits, and rhopalia). However, a commonality of mechanisms exists, such that L-Glu neurotoxicity was similar in both sexes and analogous in the different regions analysed. Nevertheless, it appears that the route of exogenous L-Glu administration influences its neurotoxic potential in vivo. In the absence of gastrointestinal tract metabolism through L-Glu administration via *s.c.* and *i.p.* routes, or when administered directly into the brain through microdialysis or stereotactic or intrastriatal injection or via brain infusion canal, extensive neurotoxicity was evident at relatively low doses. Certainly, in humans, the L-Glu neurotoxic effects on the brain could in part be mitigated by reduced circulatory levels after gastrointestinal metabolism following oral ingestion [135].

Some of the studies in this review evaluated L-Glu toxicity at prenatal [113,114] and recent postnatal time points [36,56,75,78,79,81,86,93,96,102]. The brain of the foetus or newborn animals may be more vulnerable to L-Glu damage than adult models due to an immature BBB [136]. Furthermore, the studies that used *C. elegans* and the ephyrae of *Aurelia aurita* also lack a functional BBB, allowing L-Glu to readily diffuse into the neurological system and cause neurotoxic effects [116,117].

## 5. Summary and Conclusions

In summary, excessive L-Glu intake can have pathological consequences that result in the degeneration and death of neuronal tissue. The neurotoxicity of L-Glu is mediated by multiple cellular mechanisms, including induction of redox stress and depletion of antioxidant defence, mitochondrial dyshomeostasis, excitotoxicity, neuroinflammation, altered neurotransmitter levels, and influencing of the expression and aggregation potential of key proteins involved in neurodegenerative diseases. An improved understanding and appreciation of these diverse mechanisms should enable the design of more suitable agents, such as antioxidants, that can mitigate multiple elements of acute, subacute, or even chronic neurotoxicity and neuronal damage. Furthermore, clinical, and epidemiological studies may be needed to assess the potential harm to the public from excessive intake of exogenous L-Glu.

## Figures and Tables

**Figure 1 brainsci-12-00577-f001:**
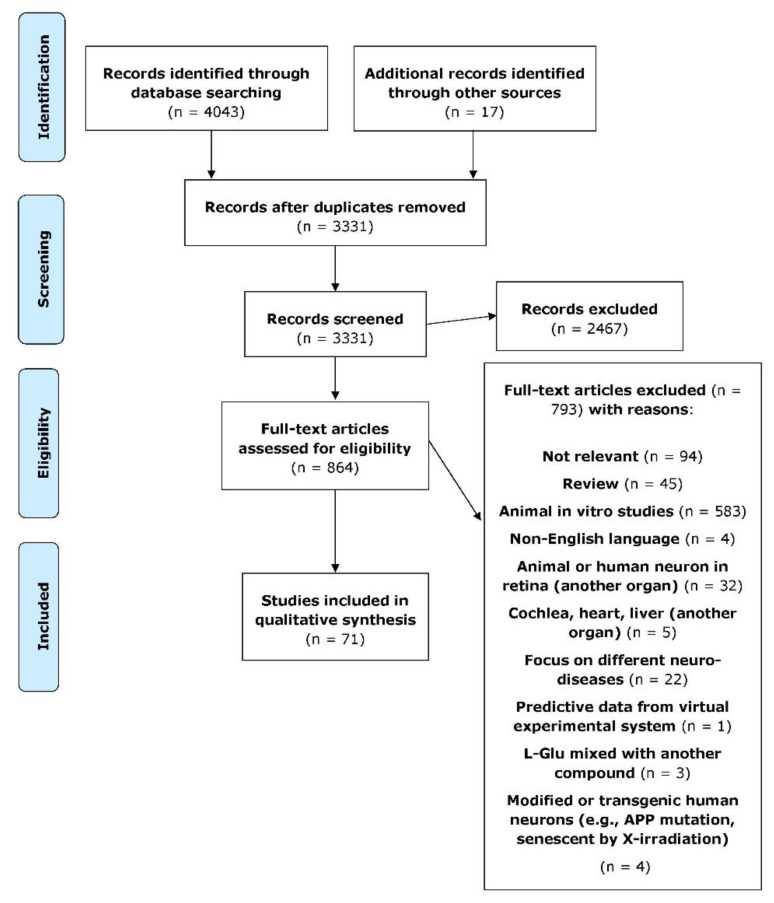
Preferred Reporting Items for Systematic Reviews and Meta-Analyses (PRISMA) flowchart illustrating the processes of data collecting and selection [50].

**Figure 2 brainsci-12-00577-f002:**
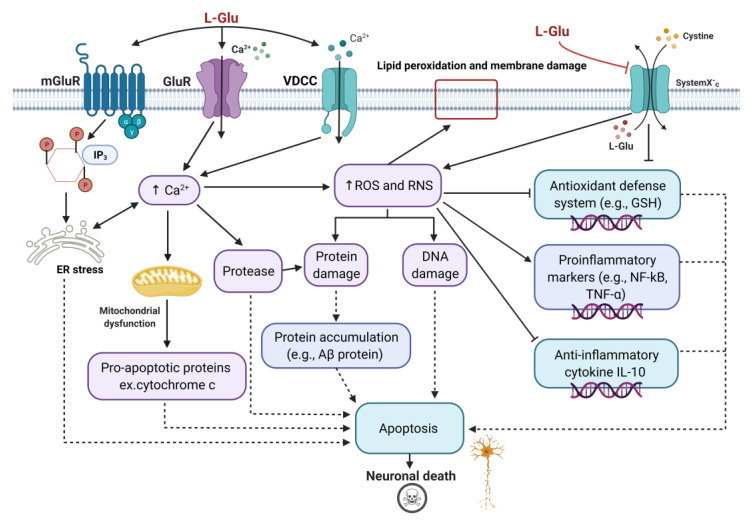
Schematic summary of the molecular mechanisms of L-Glu neurotoxicity. The binding of L-glutamate (L-Glu) to L-glutamate receptors (GluR) opens receptor channels, resulting in calcium ion (Ca^2+^) influx. L-Glu also causes voltage-dependent Ca^2+^ channels (VDCCs) to open and further Ca^2+^ influx. L-Glu activation of metabotropic glutamate receptors (mGluRs) results in increased inositol triphosphate (IP_3_) synthesis, triggering Ca^2+^ release from endoplasmic reticulum (ER) stores. Collectively, a pathological level of Ca^2+^ ions triggers ER impairment and Ca^2+^-dependent protease activation, which contributes to cellular protein damage and mitochondrial dysfunction, and neuronal apoptosis. High Ca^2+^ accumulation in neurons also leads to the generation of reactive oxygen species (ROS) and reactive nitrogen species (RNS). L-Glu inhibits the glutamate/cystine antiporter system Xc- (system Xc-), resulting in cystine depletion, an essential element for the production of the cellular antioxidant, glutathione (GSH). This impairs the endogenous antioxidant defence system and further induces redox stress. ROS and RNS cause lipid peroxidation, protein, and deoxyribonucleic acid (DNA) damage and induce the production of markers of inflammation such as nuclear factor kappa-light-chain-enhancer of activated B cell (NF-kB) activation, production of tumour necrosis factor-α (TNF-α), and inhibition of the production of the anti-inflammatory cytokine, interleukin 10 (IL-10), which collectively contributes to neuroinflammation and neuronal death. Protein modification and damage can also result in the accumulation of toxic proteins, such as amyloid beta (Aβ).

**Figure 3 brainsci-12-00577-f003:**
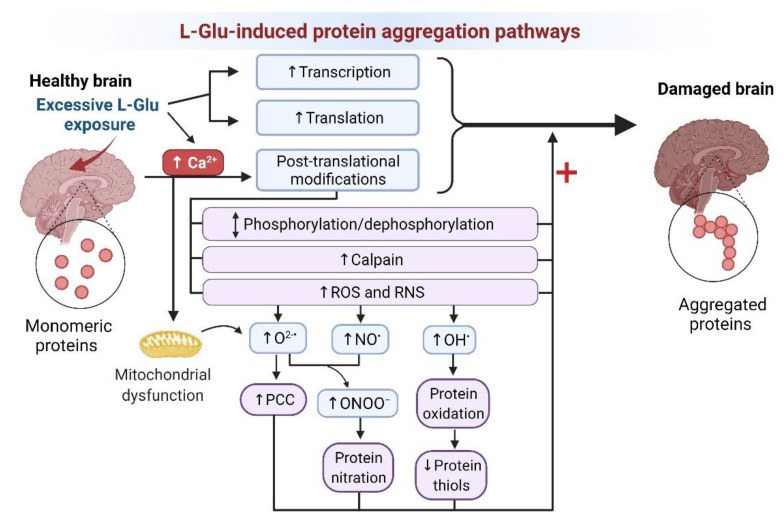
L-Glu induction of neurotoxic protein aggregation mechanisms reported from in vitro and in vivo model studies. Excessive L-Glu exposure can potentially induce protein aggregation in neurons via an influence on transcription, translation, or protein post-translation modifications (PTMs). High L-Glu in the synaptic cleft induces excitotoxicity, resulting in high Ca^2+^ influx into neurons, and this triggers protein PTMs. PTMs arise via the activation of kinases and/or phosphatases affecting the levels of protein phosphorylation/dephosphorylation, calpain activation, or increased oxidative stress (ROS or RNS production). Excessive Ca^2+^ also causes mitochondrial dysfunction, resulting in ROS leakage, including O^2−•^, which causes protein oxidation and protein carbonylation. O^2−•^ ions also interact with NO produced by nitric oxide synthase to produce reactive nitrogen species, such as ONOO^−^, which covalently modify proteins via protein nitration. L-Glu also increases the production of OH^•^, which can also cause protein oxidation and oxidation of protein thiols. Collectively, these PTMs could alter protein conformation and promote misfolding and protein aggregation. Abbreviations: Ca^2+^, calcium ions; NO^•^, nitric oxide; O^2−•^, superoxide; OH^•^, hydroxyl radical; ONOO^−^, peroxynitrite; PCC, protein carbonyl content; RNS, reactive nitrogen species; ROS, reactive oxygen species.

**Figure 4 brainsci-12-00577-f004:**
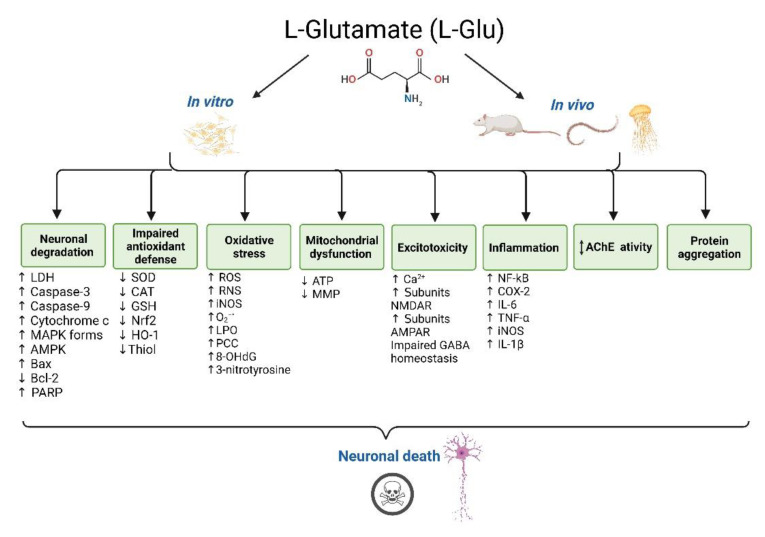
Schematic summary of the shared neurotoxicity mechanisms of L-Glu reported from in vitro and in vivo model studies. Abbreviations: AMPAR, α-amino-3-hydroxy-5-methyl-4-isoxazolepropionic acid receptor; AMPK, 5′ AMP-activated protein kinase; ATP, adenosine diphosphate; Bax, Bcl-2-associated X protein; Bcl-2, B-cell lymphoma-2; Ca^2+^, calcium ion; CAT, catalase; COX-2, cyclooxygenase-2; GABA, gamma-aminobutyric acid; GSH, glutathione; HO-1, heme oxygenase-1; IL-1β, interleukin 1 beta; IL-6, interleukin 6; iNOS, nitric oxide synthase; LDH, lactate dehydrogenase; LPO, lipid peroxidation; MAPK, mitogen-activated protein kinase; MMP, mitochondrial membrane potential; NF-κB, nuclear factor kappa-light-chain-enhancer of activated B cells; NMDAR, N-methyl-D-aspartate receptor; Nrf2, nuclear factor erythroid 2-related factor 2; O_2_^−^**^•^**, superoxide; 8-OHdG, 8-hydroxy-2′-deoxyguanosine; PARP, poly (adenosine diphosphate (ADP)-ribose) polymerase; PCC, protein carbonyl content; RNS, reactive nitrogen species; ROS, reactive oxygen species; SOD, superoxide dismutase; TNF-α, tumour necrosis factor-α.

**Table 1 brainsci-12-00577-t001:** L-glutamate in vitro studies’ outcome summary.

Study Reference	In Vitro Model	L-Glu Treatment and Duration	Study Outcomes	**Level of Significance**
**Hu et al. (2012)** [51]	SH-SY5Y undifferentiated	8 mM; 0.5 h or 12 h	↓ cell viability ↑ LDH levels Morphological alterations ↑ apoptosis ↑ Bax expression ↓ Bcl-2 expression	(*p* < 0.01) (*p* < 0.01) ND ND (*p* < 0.01) (*p* < 0.01)
**Petroni et al. (2013)** [52]	SH-SY5Y undifferentiated	1 mM; 6 h or 24 h	↓ cell viability ↑ tau protein phosphorylation	(*p* < 0.05) NS
**Jeong et al. (2014)** [53]	SH-SY5Y undifferentiated	0.01–6 mM (MSG); 24 h	↓ cell viability (5 and 6 mM) Morphological changes: pyknosis, nuclear condensation, and cytoplasmic shrinkage	(*p* < 0.01–0.001) ND
**Nampoothiri et al. (2014)** [54]	SH-SY5Y undifferentiated and differentiated	5–80 mM; 48 h	↓ cell viability	Undifferentiated (*p* < 0.001) Differentiated (*p* < 0.05–0.001)
	SH-SY5Y differentiated	20 mM; 48 h	↓ cell viability ↑ apoptosis ↓ neurite length ↑ ROS	(*p* < 0.05) (*p* < 0.01) (*p* < 0.001) (*p* < 0.001)
**Brizi et al. (2016)** [55]	SH-SY5Y undifferentiated	1–100 mM; 24 h	↓ cell viability ↑ apoptosis (50 mM) ↓ growth (50 and 80 mM) ↑ ROS (50 mM) Morphological alteration in neurons and nuclear material	ND (*p* < 0.01) (*p* < 0.01), (*p* < 0.001) ND ND
**Shah et al. (2016)** [56]	SH-SY5Y undifferentiated	10–30 mM; 3 h	↓ cell viability	(*p* < 0.05–0.001)
		30 mM; 3 h	↑ apoptosis ↑ p-AMPK protein ↓ Nrf2 protein ↓ HO-1 protein ↑ ROS ↑ p-NF-_k_B protein ↑ COX-2 protein ↑ caspase-3 protein	(*p* < 0.001) (*p* < 0.001) (*p* < 0.01) (*p* < 0.01) (*p* < 0.001) *(p* < 0.001) (*p* < 0.001) (*p* < 0.001)
**Sun et al. (2016)** [57]	SH-SY5Y undifferentiated	10–50 mM; 1, 2, 4, 6 or 8 h	↓ cell viability ↑ apoptosis and necrosis ↑ Ca^2+^ ↓ MMP (10 or 15 mM) ↑ MMP (25 or 50 mM) ↑ RIP kinase 1 protein	(*p* < 0.05) (*p* < 0.05) (*p* < 0.05) (*p* < 0.05) (*p* < 0.05) (*p* < 0.05)
**Zhu et al. (2016)** [58]	SH-SY5Y undifferentiated	5–40 mM (MSG); 24 h	↓ cell viability	(*p* < 0.05–0.01)
		20 mM (MSG); 24 h	↑ apoptosis ↑ ROS ↑ Bax protein ↓ Bcl-2 protein ↓ MMP ↓ cytosolic cytochrome c protein ↑ mitochondrial cytochrome c protein ↑ cleaved caspase-9 protein ↑ cleaved caspase-3 protein	(*p* < 0.01) (*p* < 0.01) ND ND (*p* < 0.01) ND ND ND ND
**Li et al. (2017)** [59]	SH-SY5Y undifferentiated	10 mM; 24 h	↓ cell viability ↑ LDH ↑ ROS ↑ PCC ↑ LPO ↓ SOD ↓ CAT ↓ MMP ↓ ATP ↑ mitochondrial PCC ↑ apoptosis ↑ Bax protein ↑ cleaved caspase-3 protein ↓ Bcl-2 protein ↑ p-MAPKs protein	ND ND (*p* < 0.05) (*p* < 0.05) (*p* < 0.05) (*p* < 0.05) (*p* < 0.05) ND ND ND ND ND ND ND ND
**Bharate et al. (2018)** [60]	SH-SY5Y differentiated	250 µM; 24 h	↓ cell viability	ND
**de Oliveira et al. (2019)** [61]	SH-SY5Y undifferentiated	10–80 mM; 24 h	↓ cell viability (40–80 mM)	(*p* < 0.05)
		80 mM; 24 or 6 h	↑ cleaved PARP level ↑ DNA fragmentation ↑ LPO ↑ protein nitration ↑ PCC ↓ protein thiol ↑ 8-oxo-dG level ↓ MMP ↓ ATP ↑ MC I and V activities ↓ mitochondrial enzyme activities ↑ Bax protein ↑ cytosolic cytochrome c content ↓ mitochondrial cytochrome c content ↑ caspase-9 activity ↑ caspase-3 activity ↑ O_2_^−•^ ↑ NO^•^ ↑ ROS	(*p* < 0.05) (*p* < 0.05) (*p* < 0.05) (*p* < 0.05) (*p* < 0.05) (*p* < 0.05) (*p* < 0.05) (*p* < 0.05) (*p* < 0.05) (*p* < 0.05) (*p* < 0.05) (*p* < 0.05) (*p* < 0.05) (*p* < 0.05) (*p* < 0.05) (*p* < 0.05) (*p* < 0.05) (*p* < 0.05) (*p* < 0.05)
**Lee et al. (2019)** [62]	SH-SY5Y undifferentiated	12.5–100 mM; 3 h	↓ cell viability Nuclear condensation ↑ DNA fragmentation	(*p* < 0.01) ND (*p* < 0.01)
	SH-SY5Y differentiated	100 mM; 3 h	↑ AChE activity	(*p* < 0.01)
	SH-SY5Y undifferentiated		↓ GSH level ↓ SOD protein ↓ GPx protein ↑ pp38 protein ↑ Bax protein ↑ cleaved caspase-3 protein ↑ cleaved PARP protein ↓ Bcl-2 protein	(*p* < 0.01) (*p* < 0.01) (*p* < 0.01) (*p* < 0.01) (*p* < 0.01) (*p* < 0.01) (*p* < 0.01) (*p* < 0.01)
**Xin et al. (2019)** [63]	SH-SY5Y undifferentiated	0.1–100 mM; 12 h	↓ cell viability ↑ ROS	ND ND
**Yang et al. (2019)** [64]	SH-SY5Y undifferentiated	10 mM; 24 h	↓ cell viability ↑ LDH ↑ ROS ↑ LPO ↓ SOD, GPx activities, and GSH level ↓ MMP and ATP ↑ Ca^2+^ ↑ CHOP, GRP78 proteins, and caspase-4 activity ↑ NLRP3 protein ↑ IL-1β and IL-6 ↑ Bax/Bcl-2 ratio ↑ cleaved caspase-1 and caspase-3 proteins ↑ p-MAPKs protein ↑ apoptosis	ND ND ND ND ND ND ND ND ND ND ND ND ND ND
**Yuksel et al. (2019)** [65]	SH-SY5Y undifferentiated	80 mM (MSG); 24, 48 or 72 h	↑ cell toxicity ↑ LPO ↓ SOD activity ↓ GSH level ↑ TNF-α ↑ caspase 3 and caspase 9 mRNA	ND (*p* < 0.001) (*p* < 0.05) (*p* < 0.001) (*p* < 0.001) (*p* < 0.001)
**Bebitoglu et al. (2020)** [66]	SH-SY5Y undifferentiated	1–50 mM; 3 h or 24 h	↓ cell viability (15-50 mM) ↓ CAT ↓ SOD ↑ H_2_O_2_ ↑ LPO Morphological alteration	(*p* < 0.05–0.0001) (*p* < 0.05) NS (*p* < 0.0001) (*p* < 0.0001) ND
**Fallarini et al. (2009)** [67]	Differentiated SK-N-BE(2)	1 mM; 24 h	↑ LDH ↑ Ca^2+^ ↑ c-fos and c-jun mRNA	(*p* < 0.01) (*p* < 0.01) (*p* < 0.01)
**Kataria et al. (2012)** [68]	Differentiated IMR-32 human neuroblastoma	0.06–10 mM; 24 h	Morphological changes, cell shrinkage, and rounding ↓ cell viability ↑ LDH ↓ NF200 mRNA and protein (0.25 and 0.5 mM) ↑ HSP70 mRNA and protein (0.25 and 0.5 mM) ↑ PSA-NCAM expression (0.5 mM) ↑ PST mRNA (0.25 and 0.5 mM)	ND (*p* < 0.05) (*p* < 0.05) (*p* < 0.05) (*p* < 0.05) (*p* < 0.05) (*p* < 0.05)
**Occhiuto et al. (2008)** [69]	Differentiated HCN-1A cell line	0.01–5 mM; 24 h	↓ cell viability	ND
		0.1 mM; 24 h	↓ cell viability ↑ LDH Neuron morphological alteration	(*p* < 0.01) (*p* < 0.05) ND
**Palumbo et al. (2012)** [70]	Differentiated HCN-1A cell line	0.1 mM; 24 h	↑ LDH ↓ cell viability Neuron morphological alteration ↑ Ca^2+^	(*p* < 0.05) (*p* < 0.05) ND (*p* < 0.01)
**Gupta et al. (2013)** [71]	Human embryonic stem cell (HESC) line H9	20-200 µM; 24 h	↑ neuronal death (20–80 µM) ↑ Ca^2+^ influx (200 µM)	ND (*p* < 0.001)
**Yon et al. (2018)** [72]	Human neural stem cell (NSC) culture HB1.F3	0.8–50 mM (MSG); 2 h	↑ LDH	ND
		2.5 mM (MSG); 2 h	↑ LDH ↑ NF-κB mRNA ↑ TNF-α mRNA ↑ IL-6 mRNA ↑ iNOS mRNA and protein ↑ COX-2 mRNA and protein ↑ TGF-β protein ↑ HMGB1 protein	(*p* < 0.05) (*p* < 0.05) (*p* < 0.05) (*p* < 0.05) (*p* < 0.05) (*p* < 0.05) (*p* < 0.05) (*p* < 0.05)
**Chao and Hu (1994)** [73]	Human fetal brain tissue	1.6–5000 µM; 6 d	↑ LDH ↓ GABA uptake (as marker of GABAergic neuron integrity)	ND ND
**de Vera et al. (2008)** [74]	Human fetal cortical brain tissue 14–42 DIV	1–10 mM; 24 h	↑ neuronal death (5 mM at 26, 32, and 42 DIV) ↑ swelling of astrocyte nuclei (5 mM at 32 DIV)	(*p* < 0.05) (*p* < 0.05)

Abbreviations: AChE, acetylcholinesterase; AMPAR, α-amino-3-hydroxy-5-methyl-4-isoxazolepropionic acid receptor; ATP, adenosine triphosphate; Bax, Bcl-2-associated X protein; Bcl-2, B-cell lymphoma-2; Ca^2+^, calcium ions; CAT, catalase; CHOP, CCAAT/enhancer-binding protein homologous protein; COX-2, cyclooxygenase-2; DIV, day in vitro; DNA, deoxyribonucleic acid; FJB, fluoro jade B; GPx, glutathione peroxidase; GRP78, glucose regulatory protein 78; GSH, glutathione; GSSG, oxidised glutathione; GST, glutathione-S-transferase; GABA, gamma-aminobutyric acid; HCN-1A line, human cortical neuronal cells; HMGB1, high-mobility group box 1; HO-1, heme oxygenase-1; H_2_O_2_, hydrogen peroxide; HSP70, 70 kDa heat shock protein; IL-1β, interleukin 1β; IL-6, interleukin 6; iNOS, inducible nitric oxide synthase; LDH, lactate dehydrogenase; L-Glu, L-glutamate; LPO, lipid peroxidation; MC I, mitochondrial complex I; MC V, mitochondrial complex V; MMP, mitochondrial membrane potential; mRNA, messenger RNA; MSG, monosodium glutamate; ND, not determined; NF200, neurofilament protein 200; NLRP3, Nod-like receptor protein 3; NMDAR, N-methyl-D-aspartate receptor; NO^−^, nitrite; NO^•^, nitric oxide radical; Nrf2, nuclear factor erythroid 2-related factor; NS, not significant; O_2_^−^**^•^**, superoxide radical; 8-oxo-dG, 8-oxo-2′-deoxyguanosine; p38 MAPK, p38 mitogen-activated protein kinase; p-AMPK, phosphorylated AMP-activated protein kinase; PARP, poly (adenosine diphosphate (ADP)-ribose) polymerase; PCC, protein carbonyl content; p-MAPKs, phosphorylated mitogen-activated protein kinases; p-NF-_k_B, phosphorylated nuclear factor kappa-light-chain-enhancer of activated B cells; pp38, phosphorylated-p38 mitogen-activated protein kinase; PSA-NCAM, polysialylated neural cell adhesion molecule; PST, polysialyltransferase; RIP kinase 1, receptor-interacting protein kinase 1; ROS, reactive oxygen species; SH-SY5Y, neuroblastoma cell line; SK-N-BE(2), human neuroblastoma cell lines; SOD, superoxide dismutase; TGF-β, transforming growth factor beta; TNF-α, tumour necrosis factor-α. Note: Malondialdehyde (MDA) level used as an index of lipid peroxidation (LPO).

**Table 2 brainsci-12-00577-t002:** L-glutamate in vivo studies’ outcome summary.

Reference	Species and Strain, Size of L-Glu Treatment Group	L-Glu Treatment and Duration (Dose, Route of Drug Application)	Study Outcomes	Level of Significance
**Babu et al. (1994)** [75]	Wistar rats (n = 6)	MSG; 4 mg/g, *s.c.*; 10 days postnatally (PD 1–10)	↑ LPO ↑ CAT activity ↓ sulfhydryl levels	(*p* < 0.01) (*p* < 0.025) (*p* < 0.05)
**Ferger et al. (1998)** [76]	Albino Wistar rats (n = 5) adults male (350 g)	L-Glu 50 mM, striatum microdialysis	↑ OH^•^	ND
**Singh et al. (2003)** [77]	Wistar rats (n = 3) male (160–180 g) 3–4 months old	MSG 4 mg/g/day, *i.p.*; 6 consecutive days	↓ Mn-SOD activity ↓ CAT activity (mitochondrial) ↓ GSH content ↑ GPx content ↑ LPO ↑ uric acid	(*p* < 0.001) (*p* < 0.01–0.001) (*p* < 0.05–0.001) (*p* < 0.05–0.001) (*p* < 0.02–0.001) (*p* < 0.05–0.01)
**Rivera-Cervantes et al. (2004)** [78]	Wistar rats (n = 6)	MSG; 4 mg/g, *s.c.*; PD 1, 3, 5, 7	↑ neurons’ histologic changes and degeneration ↑ Gliosis ↑ NR1 mRNA subunit of NMDAR ↑ GluR2 mRNA subunit of AMPAR ↑ ATF2 ^pp^ protein (for p38 MAPK activation)	(*p* < 0.001) ND (*p* < 0.001) (*p* < 0.001) (*p* < 0.001)
**Chaparro-Huerta et al. (2005)** [79]	Wistar rats (n = 8)	MSG: 4 mg/g, *s.c.*; PD 1, 3, 5, 7	↓ glial size and processes ↑ apoptosis ↑ TNF-α mRNA and protein ↑ IL-1ß mRNA and protein ↑ IL-6 mRNA and protein	ND (*p* < 0.001) (*p* < 0.05, and *p* < 0.001) (*p* < 0.001) (*p* < 0.001)
**Mejía-Toiber et al. (2006)** [80]	Wistar rats (n = 3–9) male (250–320 g)	1 µmol/µL, intrastriatal injection; 0.5 µL/min for 2 min	↑ striatal lesions ↓ ATP level (after 6 h but not at 3 h) ↑ LPO level (after 5 or 24 h)	(*p* < 0.0005) ND (*p* < 0.05)
**Segura Torres et al. (2006)** [81]	Wistar rats (n = 5)	MSG; 4 mg/g, *s.c.*; PD 1, 3, 5, 7	↑ GluR2 protein subunit of AMPAR at PD 8 ↓ GluR2 protein subunit of AMPAR at PD 14 ↑ REST mRNA at PD 8 and 14 ↑ Fas-L and Bcl-2 mRNA at PD 8	ND ND (*p* ≤ 0.01–0.001) (*p* ≤ 0.01)
**Chaparro-Huerta et al. (2008)** [36]	Wistar rats (n = 8)	MSG; 4 mg/g, *s.c.*; PD 1, 3, 5, 7	↑ apoptosis ↑ nuclear material condensed ↑ TNF-α mRNA ↑ IL-1ß mRNA ↑ IL-6 mRNA	(*p* < 0.05, *p* < 0.001) ND (*p* < 0.05–*p* < 0.001) (*p* < 0.001) (*p* < 0.001)
**Del Río and Massieu (2008)** [82]	Wistar rats (n = 4–7) male (250–300 g)	1 µL (500 nmoles), intrastriatal injections; rate of 0.5 µL/min	↑ brain lesions ↑ calpain activation protein	ND (*p* ≤ 0.05)
**Hashem et al. (2012)** [83]	Albino rats (n = 10) adult male (150–200 g) 3–6 months old	MSG; 3 g/kg/day, *p.o.*; 14 days	Neurons’ morphological alterations Darkly stained cytoplasm of Purkinje cells Shrunken darkly stained nuclei ↑ neurons degeneration ↑ inflammatory cells ↑ GFAP immunoreactivity in the astrocytes of granular layer	ND
**Shivasharan et al. (2013)** [27]	Wistar rats (n = 6) adult female (190–220 g)	MSG; 2 g/kg/day, *i.p.*; 7 days	↓ locomotor activity Decreased hippocampus layer, darkly stained shrunken cells, and mildly separated interconnected neuropil fibres ↑ NO^−^ ↑ LPO level ↓ GSH level ↓ GST activity ↓ CAT activity ↓ total thiols level	(*p* < 0.001) ND (*p* < 0.001) (*p* < 0.001) (*p* < 0.001) (*p* < 0.001) (*p* < 0.05) (*p* < 0.01)
**Swamy et al. (2013)** [84]	Wistar albino rats (n = 6) (50–200 g) of either sex	MSG 2 g/kg/day, *i.p.*; 7 days	↑ behavioural alterations and reduced locomotor activity Marked cerebral oedema, neuronal eosinophilia, nuclear pyknosis, and neuronal karyorrhexis ↑ Ca^2+^ ↑ Na^+^ ↓ K^+^ ↓ GABA ↓ GSH level ↓ SOD activity ↓ CAT activity ↑ LPO levels	(*p* < 0.05) ND (*p* < 0.05) (*p* < 0.05) (*p* < 0.05) (*p* < 0.05) (*p* < 0.05) (*p* < 0.05) (*p* < 0.05) (*p* < 0.05)
**Dief et al. (2014)** [33]	Wistar rats (n = 6) male (40–60 g) 5 weeks old	MSG; 2 g/kg/day, *p.o.*; 10 consecutive days MSG; 4 g/kg/day, *s.c.*; 10 alternate days	↑ anxiety behaviour ↓ working memory ↓ AMPK protein ↑ Fas-L protein ↑ Aβ (1-42)	(*p* < 0.05) (*p* < 0.05) (*p* < 0.05) (*p* < 0.05) (*p* < 0.05)
**Thonda et al. (2014)** [85]	Wistar rats (n = 6) adult female (180–220 g)	MSG 2 g/kg/day, *i.p.*; 7 days	↓ locomotor activity ↓ grip strength ↓ memory retention ↓ motor coordination and body balance Hippocampal pyramidal cells’ degeneration with intact neuropil fibres ↓ body weight ↑ LPO levels ↓ CAT activity ↓ SOD activity ↓ GSH level	(*p* < 0.001) (*p* < 0.01) (*p* < 0.01) (*p* < 0.001) ND (*p* < 0.01) (*p* < 0.001) (*p* < 0.01) (*p* < 0.001) (*p* < 0.01)
**Rivera-Cervantes et al. (2015)** [86]	Wistar rats (n = 4–5)	MSG; 4 mg/g, *s.c.*; PD 1, 3, 5, 7	↑ alterations and loss in the hippocampal neurons at PD 8, 10, 12, and 14 ↑ TUNEL-positive cells at PD 8, 10, and 14 ↑ NMDAR subunit NR1 mRNA at PD 10, 12, and 14 ↑ AMPAR subunit GluR1 mRNA at PD 12 and 14 ↓ AMPAR subunit GluR2 mRNA and protein at PD 10 and 14 ↑ NRSF mRNA at PD 8 and 14 ↑ ATF2^pp^ protein	(*p* < 0.01) (*p* < 0.001) (*p* < 0.001) (*p* < 0.001) (*p* < 0.001) (*p* < 0.01) (*p* < 0.001)
**Khalil and Khedr (2016)** [87]	Wistar rats (n = 8) male (120–150 g) 12 weeks old	MSG; 4 mg/kg/day, *p.o.*; 4 weeks	↑ L-Glu level ↑ AChE activity ↑ TNF-α level ↑ NMDA2B mRNA ↑ mGluR5 mRNA	(*p* < 0.01) (*p* < 0.001) (*p* < 0.001) (*p* < 0.001) NS
**Sadek et al. (2016)** [88]	Albino Wistar rats (n = 8) male (130–160 g) 2 months old	MSG; 5 mg/kg/day, *s.c.*; 4 weeks	↑ LDH ↑ Na^+^ ↓ K^+^ ↑ LPO level ↑ GST activity and mRNA ↑ CAT activity and mRNA ↑ SOD activity ↓ GSH level ↑ AChE activity ↑ Bax mRNA ↓ Bcl-2 mRNA ↑ Serum ChE level ↑ CPK activity ↑ CPK-BB activity	(*p* < 0.05) (*p* < 0.05) (*p* < 0.05) (*p* < 0.05) (*p* < 0.05) (*p* < 0.05) (*p* < 0.05) (*p* < 0.05) (*p* < 0.05) (*p* < 0.05) (*p* < 0.05) (*p* < 0.05) (*p* < 0.05) (*p* < 0.05)
**Hussein et al. (2017)** [89]	Albino rats (n = 6) male (100–130 g)	MSG; 100 mg/kg/day, *p.o.*; 2 months	Pathological damage to brain tissue ↑ LPO level ↑ NO^•^ ↓ SOD activity ↓ CAT activity ↓ GSH level ↑ Aβ (1-42) ↓ AChE activity ↑ serotonin level ↑ dopamine level ↑ L-Glu level ↑ Ca^2+^ ↑ Na^+^ ↓ K^+^ ↑ 8-OHdG in the brain DNA	ND (*p* < 0.01) (*p* < 0.001) (*p* < 0.001) (*p* < 0.001) (*p* < 0.01) (*p* < 0.001) (*p* < 0.01) (*p* < 0.001) (*p* < 0.001) (*p* < 0.001) (*p* < 0.05) (*p* < 0.05) (*p* < 0.05) (*p* < 0.01)
**Abdel Moneim et al. (2018)** [90]	Albino rats (n = 20) male (45–70 g) 5–6 weeks old	High; MSG 1.66 g/kg/day, *p.o.*; 30 days Low; MSG 0.83 g/kg/day, *p.o.*; 30 days	↓ cognitive functions ↓ serotonin level (high-dose MSG)	(*p* < 0.001) (*p* < 0.001)
**Fouad et al. (2018)** [91]	Albino rats (n = 12) adult male (250–300 g) 2 months old	2 g/kg/day, *i.p.*; 7 days	↓ spontaneous alternation behaviour (spatial working memory) ↑ MEL in the MWM ↓ time spent in target quadrant (MWM) ↑ cytochrome c mRNA ↑ caspase-3 level ↑ LDH ↑ Aβ (1-42)	(*p* < 0.05) (*p* < 0.05) (*p* < 0.05) (*p* < 0.05) (*p* < 0.05) (*p* < 0.05) (*p* < 0.05)
**Hazzaa et al. (2020)** [92]	Wistar albino rats (n = 10) male (40 g) 1 month old	MSG 4 g/kg/day, *i.p.*; 7 days	↓ locomotor activity ↓ spatial memory ↑ morphological alteration in hippocampus neurons ↑ LPO level ↑ caspase-3 protein ↓ SOD activity ↑ GFAP protein ↓ Ki-67 protein ↑ calretinin protein	(*p* < 0.001) (*p* < 0.05–0.001) (*p* < 0.05) (*p* < 0.05) ND (*p* < 0.05) (*p* < 0.05) (*p* < 0.05) (*p* < 0.05)
**Greene and Greenamyre (1995)** [94]	Sprague–Dawley rats (n = 5) male (200–250 g)	L-Glu 0.3 M intrastriatal injection; 2 µL of solution at 0.5 µL/min (0.6 µmoles)	↑ lesion	(*p* < 0.01)
**Yang et al. (1998)** [95]	Sprague–Dawley rats (n = 10) male (280–350 g)	1.5 or 15 mM L-Glu, cortex microdialysis; 2 μL/min 20–180 min	↑ LPO at 1.5 and 15 mM	ND
**Bodnár et al. (2001)** [96]	Sprague–Dawley rats (n = 7–9) of both sexes	MSG; 4 mg/g, *s.c.;* PD 2, 4, 6, 8, 10	↑ degradation of TH-positive (dopaminergic) neurons of hypothalamic arcuate nucleus	(*p* < 0.05)
**Kumar and Babu (2010)** [97]	Sprague–Dawley rats (n = 6) male (300–350 g) 3 months old	1 µmole/1 µL, cerebral cortex injection	↓ pyramidal neurons’ size ↑ condensed nuclei ↑ Ca^2+^ ↑ LPO level ↑ ROS ↓ SOD activity ↓ CAT activity ↓ GSH level ↓ GR activity ↑ TNF-α level ↑ IFN-ɣ level ↑ NO^•^ ↓ MMP	ND (*p* < 0.001) (*p* < 0.001) (*p* < 0.001) (*p* < 0.001) (*p* < 0.001) (*p* < 0.001) (*p* < 0.05) (*p* < 0.001) (*p* < 0.001) (*p* < 0.05) (*p* < 0.001) (*p* < 0.001)
**Kumar et al. (2010)** [98]	Sprague–Dawley rats (n = 6) male (300–350 g) 3 months old	1 µmole/1 µL, cerebral cortex injection	↓ pyramidal neurons’ size ↑ condensed nuclei ↑ ROS ↑ ONOO^−^ ↓ MMP ↓ GSH ↑ Ca^2+^ ↑ nNOS mRNA ↑ iNOS mRNA ↑ caspase-3 mRNA ↑ caspase-9 mRNA ↓ Bcl-2 mRNA ↑ Bax mRNA ↓ Bcl-2/Bax ratio mRNA	ND (*p* < 0.001) (*p* < 0.001) (*p* < 0.001) (*p* < 0.001) (*p* < 0.001) (*p* < 0.001) (*p* < 0.001) (*p* < 0.001) (*p* < 0.001) (*p* < 0.001) ND ND (*p* < 0.001)
**Nagesh Babu et al. (2011)** [99]	Sprague–Dawley rats (n = 6) male (300–350 g) 3 months old	1 µmole/1 µL, cerebral cortex injection	↑ LPO level ↑ ROS ↓ SOD activity ↓ CAT activity ↓ GSH level ↓ GR activity ↑ TNF-α level ↑ IFN-ɣ level ↑ NO^•^ ↓ MMP ↑ Ca^2+^ ↑ caspase-3 mRNA ↑ caspase-9 mRNA ↑ iNOS mRNA ↑ nNOS mRNA	(*p* < 0.001) (*p* < 0.001) (*p* < 0.05) (*p* < 0.05) (*p* < 0.05) (*p* < 0.05) (*p* < 0.001) (*p* < 0.001) (*p* < 0.001) (*p* < 0.001) (*p* < 0.001) ND ND ND ND
**Morales and Rodriguez (2012)** [100]	Sprague–Dawley rats (n = 5, 7, 8) male (300–350 g)	L-Glu in Ringer solution of 2 µL/min, striatum microdialysis; 60 min	↑ astrogliosis ↑ L-Glu level ↑ alanine level ↓ glutamine level	(*p* < 0.001) (*p* < 0.001) (*p* < 0.001) (*p* < 0.001)
**Kim et al. (2013)** [101]	Sprague–Dawley rats (n = 6) male (300–350 g) 3 months old	1 µmol/1 µL, cerebral cortex injection	↑ LPO level ↑ Ca^2+^ ↑ ROS ↓ GSH level ↓ SOD activity ↓ CAT activity ↓ GPx level ↓ GR level ↑ TNF-α level ↑ IFN-ɣ level ↑ NO^−^ ↑ NADPH oxidase activity ↑ nNOS mRNA ↓ MMP ↑ p-ERK1/2 mRNA ↑ caspase-3 mRNA	ND ND ND ND ND ND ND ND ND ND ND ND ND ND ND ND
**Shah et al. (2015)** [102]	Sprague–Dawley male rats (n = 10) male (18 g) pups at PD 7	5 mg/kg, *s.c.*; 4 or 12 h, PD 7 10 mg/kg, *s.c.*; 4 or 12 h, PD 7	↑ L-Glu level ↑ Bax protein ↓ Bcl-2 protein ↑ Bax/Bcl-2 ratio ↑ cytochrome c protein ↑ cleaved caspase-3 protein and level ↑ p-AMPK protein (high dose) ↑ FJB positive neurons (high dose) ↑ cleaved PARP-1 protein	(*p* < 0.01–0.0001) ND ND (*p* < 0.01–0.0001) (*p* < 0.01–0.0001) (*p* < 0.01–0.0001) (*p* < 0.0001) (*p* < 0.0001) (*p* < 0.01–0.0001)
**Calis et al. (2016)** [103]	Sprague–Dawley rats (n = 7) Female (250–300 g) 3–4 months old	MSG; 2 g/kg/day, *i.p.*; 7 days	No neuron degeneration in pyramidal and granular neurons in the brain cortex No effect on SOD level ↓ LPO level	ND NS (*p* < 0.001)
**Shah et al. (2016)** [56]	Sprague–Dawley rats (n = 5) (18 g) pups at PD 7	10 mg/kg, *i.p.*; 2, 3, 4 h	↑ DNA fragmentation ↑ L-Glu ↑ AMPAR protein ↑ p-AMPK protein ↑ p-NF-_k_B protein ↓ Nrf2 protein ↑ CaMKII protein ↓ GSH level ↓GSH/GSSG ratio ↑ GFAP ↑ microglia Iba-1 ↑ ROS ↑ COX-2 protein ↑ TNF-α protein ↑ caspase-3 protein ↓ HO-1 protein	(*p* < 0.001) (*p* < 0.05–0.001) (*p* < 0.05–0.001) (*p* < 0.05–0.001) (*p* < 0.05–0.001) (*p* < 0.05–0.01) (*p* < 0.001) (*p* < 0.001) (*p* < 0.001) (*p* < 0.001) (*p* < 0.001) (*p* < 0.001) (*p* < 0.001) (*p* < 0.001) (*p* < 0.001) (*p* < 0.001)
**Yang et al. (2017)** [104]	Sprague–Dawley rats (n = 6) adult male (270–320 g)	1 M/1 μL, cerebral cortex injection	↑ TUNEL-positive cells ↑ caspase-3 protein and activity ↑ calpain protein and activity ↑ Bax protein ↓ Bcl-2 protein ↓ Bcl-2/Bax ratio ↑ ROS ↑ NO ↑ TNF-α level ↑ IFN-ɣ level ↑ IL-1β level ↑ NOX activity ↑ LPO level ↓ SOD activity ↓ GR activity ↓ GSH level ↓ CAT activity ↑ iNOS protein ↑ nNOS protein ↓ Nrf2 protein ↓ HO-1 protein ↓ GCLC protein ↓ ATP level ↓ Na^+^-K^+^-ATPase level ↓ cytochrome c oxidase activity	ND ND ND ND ND ND ND ND ND ND ND ND ND ND ND ND ND ND ND ND ND ND ND ND ND
**Firgany and Sarhan (2020)** [105]	Sprague–Dawley rats (n = 10) male (365 g) 18 months old	MSG; 4.0 g/kg/day, *s.c.*; 10 days	Remarkable morphological alteration in motoneurons and neuroglia ↑ caspase-3 activity ↑ LPO level ↑ IL-1β level ↑ IL-6 level ↑ TNF-α level ↑ IFN-ɣ level ↓ IL-10 level ↓ SOD activity ↓ CAT activity ↓ GFAP level ↑ ATF2^pp^ protein	ND (*p* < 0.001) (*p* < 0.001) (*p* < 0.001) (*p* < 0.001) (*p* < 0.001) (*p* < 0.001) (*p* < 0.001) (*p* < 0.001) NS (*p* < 0.001) (*p* < 0.001)
**Hamza et al. (2019)** [106]	Rats (n = 8) adult male (200–250 g)	MSG high; 17.5 mg/kg/day, *p.o.*; 30 days MSG low; 6 mg/kg/day, *p.o.*; 30 days	MSG high dose: Large area of haemorrhage and necrotic areas of the brain with the congested area with degeneration in some glial ↓ catecholamine level ↓ dopamine level ↓ serotonin level ↓ AChE activity ↓ thiol level ↓ SOD activity ↓ CAT activity ↓ GPx activity ↓ GSH level ↓ BDNF level ↑ COX-2 activity ↑ PGE2 level MSG low dose: Moderate area of haemorrhage and necrosis in the brains	ND (*p* ≤ 0.05) (*p* ≤ 0.05) (*p* ≤ 0.05) (*p* ≤ 0.05) (*p* ≤ 0.05) (*p* ≤ 0.05) (*p* ≤ 0.05) (*p* ≤ 0.05) (*p* ≤ 0.05) (*p* ≤ 0.05) (*p* ≤ 0.05) (*p* ≤ 0.05) ND
**Burde et al. (1971)** [93]	Wistar rat 4 day old and Swiss albino mice 10 day old (n = 2–9)	MSG; 1 mg/g, 4 mg/g, *p.o., or s.c.*; 5 h MSG; 4 mg/g, 2 mg/g, *p.o., or s.c.*; 5 h	↑ lesions in arcuate of the hypothalamus Lesions more extensive in rat than mice ↑ necrosis of neuron of arcuate Perikaryon swelling loss of cytoplasmic density and nuclear pyknosis	ND
**Onaolapo et al. (2016)** [28]	Swiss mice (n = 10) adult male (20–22 g)	MSG; 10, 20, 40, and 80 mg/kg/day, *p.o*.; 28 days L-Glu; 10 mg/kg/day, *p.o.*; 28 days	MSG: ↑ brain weight (40 and 80 mg/kg) ↑ neurons’ morphological alteration (MSG and L-Glu) ↑ glial cell number ↑ L-Glu plasma level (40 and 80 mg/kg) ↑ glutamine plasma level (40 and 80 mg/kg) ↓ SOD level (20, 40, and 80 mg/kg) ↓ CAT level (40 and 80 mg/kg) ↑ NO^•^ (all doses)	(*p* < 0.01) (*p* < 0.05) (*p* < 0.05) (*p* < 0.001) (*p* < 0.01) (*p* < 0.01) (*p* < 0.001) (*p* < 0.002)
**Mohan et al. (2017)** [107]	Swiss albino mice (n = 5 for some experiments) of either sex (18–22 g)	MSG; 1000 mg/kg/day, *p.o.*; 14 days	↓ onset of immobility delayed ↓ total immobility period ↓ brain weight Neurodegeneration characterised by deformed brain layers, pyknosis, and neuronal cell vacuolisation ↓ SOD activity ↓ CAT activity ↓ RGSH ↑ LPO level	(*p* < 0.05) (*p* < 0.05) (*p* < 0.05) ND (*p* < 0.05) (*p* < 0.05) (*p* < 0.05) (*p* < 0.05)
**Estrada-Sánchez et al. (2009)** [108]	Wild-type mice (n = 3–4) female 10–14 weeks old	500 nM/0.5 μL, intrastriatal injection; 3 and 24 h	↑ striatal lesions at 10 and 14 weeks ↑ FJB-positive degenerating neurons	(*p* ≤ 0.05) (*p* ≤ 0.05)
**Estrada-Sánchez et al. (2010)** [109]	Wild-type mice (n = 3–6) female 10 weeks old	500 nM/0.5 µL, intrastriatal injection; 24 h	↑ lesions	(*p* ≤0.05)
**Guemez-Gamboa et al. (2011)** [110]	Wild-type mice (n = 3–7)	MSG; 500 nM/0.5 µL, intrastriatal injection; rate of 0.175 µL/min	↑ microglia activation ↑ NT-positive cells (NT; index of oxidative damage) ↑ striatal lesions ↑ FJB-positive degenerating neurons ↑ ROS ↑ NADPH oxidase activity ↑ calpain protein	ND (*p* < 0.05) (*p* < 0.001) (*p* < 0.001) (*p* < 0.001) (*p* < 0.05) (*p* < 0.05)
**Wang et al. (2015)** [111]	Wild-type mice (n = 3) male 4–6 months old	10 mM at a flow rate of 1 µL/h, left lateral ventricle, brain infusion cannula; 7 days	↑ motor neurons clumping or fragmented nuclei ↑ mitochondrial fragmentation ↓ MFN2 protein ↑ cleaved caspase-3-positive neurons	ND (*p* < 0.001) (*p* < 0.05) ND
**Zou et al. (2016)** [112]	Wild-type mice (n = 6) (25–30 g) 4–5 months old	MSG; 0.25 M, 0.2 μL injected on each side of the parietal cortex	↑ lesion	(*p* < 0.01)
**Yu et al. (2006)** [113]	Kunming mice (n = 8–10) female, pregnant 7 weeks old	MSG; 1, 2, 4 g/kg/day, *i.g.*; at days 17–19 days of pregnancy	↑ hyperactivity from open field test ↓ the memory retention and Y-maze discrimination learning capacities ↑ hippocampal lesions ↑ [^3^H]-Glu uptake ↓ Bcl-2 protein ↑ caspase-3 protein	(*p* < 0.0001- 0.0004) (*p* < 0.0001) ND ND ND ND
**Ma et al. (2007)** [114]	Kunming mice (n = 11–13) adults 8 weeks old	MSG; 1, 2, 4 g/kg/day, *i.g*.; 10 days	↓ discrimination learning and memory using Y-maze test Hippocampal lesions’ intracellular oedema, degeneration and necrosis of neurons, and hyperplasia	(*p* = 0.0006) ND
**Penugonda and Ercal (2011)** [115]	CD-1 mice (n = 4) adult male (38–40 g)	2000 mg/kg/day, *p.o.*; 1 week	↓ GSH level ↑ LPO level ↑ PLA2s activity	NS (*p* < 0.05–0.005) (*p* < 0.05–0.001)
**Chu et al. (2020)** [116]	*Caenorhabditis elegans* (wild-type) nematodes (n = 30)	20 mM, animals living media; 24 h	↑ damaged locomotory ability ↑ ROS ↑ O_2_^−•^ ↓ GSH level	(*p* < 0.05) ND ND ND
**Spangenberg et al. (2004)** [117]	Ephyrae of *Aurelia aurita* (n = 2–4)	MSG; 5 mM, animals living media (artificial sea water); 1–24 h	↑ impaired pulsing and swimming motility ↓ pulsing rates ↑ Ca^2+^ ↑ ROS ↑ NO^•^	(*p* < 0.05) (*p* < 0.05) (*p* < 0.05) (*p* < 0.05) ND

Abbreviations: AChE, acetylcholinesterase; AMPAR, α-amino-3-hydroxy-5-methyl-4-isoxazolepropionic acid receptor; AMPK, 5′AMP-activated protein kinase; ATF2^pp^, activating transcription factor 2 phosphorylated; ATP, adenosine triphosphate; Aβ, amyloid beta; Bax, Bcl-2-associated X protein; Bcl-2, B-cell lymphoma-2; BDNF, brain-derived neurotrophic factor; Ca^2+^, calcium ions; CaMKII, Ca^2+^/calmodulin-dependent protein kinase II; CAT, catalase; ChE, cholinesterase; COX-2, cyclooxygenase-2; CPK, creatine phosphokinase; CPK-BB, creatine phosphokinase isoenzymes BB; DNA, deoxyribonucleic acid; Fas-L, Fas ligand; FJB, fluoro jade B; GABA, gamma-aminobutyric acid; GCLC, glutamate cysteine ligase catalytic subunit; GFAP, glial fibrillary acidic protein; GPx, glutathione peroxidase; GR, glutathione reductase; GSH, glutathione; GSSG, oxidised glutathione; GST, glutathione-S-transferase; ^3^H-GABA, [^3^H]gamma-aminobutyric acid (GABA); [^3^H]-Glu, [^3^H]-glutamate; HO-1, heme oxygenase-1; *i.g*., maternal intragastric; *i.p.*, intraperitoneal injection; Iba-1, ionised calcium-binding adaptor molecule 1; IFN-ɣ, interferon gamma; IL-1β, interleukin 1 beta; IL-10, interleukin 10; IL-6, interleukin 6; iNOS, inducible nitric oxide synthase; K^+^, potassium ions; LDH, lactate dehydrogenase; L-Glu, L-glutamate; LPO, lipid peroxidation; MEL, mean escape latency; MFN2, mitofusin 2; mGluR5, metabotropic glutamate receptor 5; MMP, mitochondrial membrane potential; Mn-SOD, manganese superoxide dismutase; mRNA, messenger RNA; MSG, monosodium glutamate; MWM, Morris water maze; Na^+^, sodium ions; Na^+^-K^+^-ATPase, sodium potassium adenosine triphosphatase; ND, not determined; NMDA2B, N-methyl-D-aspartate receptor 2B; NMDAR, N-methyl-D-aspartate receptor; nNOS, neuronal nitric oxide synthase; NO^−^, nitrite; NO^•^, nitric oxide; Nrf2, nuclear factor E2-related factor 2; NRSF, neuron-restrictive silencer factor; NS, not significant; NT, nitrosylated proteins; O_2_^−^**^•^**, superoxide; OH^•^, hydroxyl radicals; 8-OHdG, 8-hydroxy-2′-deoxyguanosine; ONOO^−^, peroxynitrites; p-AMPK, phosphorylate AMP-activated protein kinase; PARP-1, (ADP ribose) polymerase-1; PD, postnatal day; p-ERK 1/2, phospho-extracellular signal-regulated kinase; PGE2, prostaglandin E2; PLA2s, phospholipases A2; p38 MAPK, p38 mitogen-activated protein kinases; p-NF-_k_B, phosphorylate nuclear factor kappa-light-chain-enhancer of activated B cells; *p.o.*, orally; REST, RE1-silencing transcription factor; RGSH, reduced glutathione; ROS, reactive oxygen species; *s.c.*, subcutaneous injection; SOD, superoxide dismutase; TH, tyrosine hydroxylase; TNF-α, tumour necrosis factor alpha; TUNEL, terminal deoxynucleotidyl transferase (dUTP) nick end labeling. Note: malondialdehyde (MDA) level is an index of lipid peroxidation (LPO).

## Data Availability

Data supporting the results are available on request from the first author (M.N.A.)

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
