# Peer review of "Is L-Glutamate Toxic to Neurons and Thereby Contributes to Neuronal Loss and Neurodegeneration? A Systematic Review"

_brainsci, 2022, doi:10.3390/brainsci12050577_

Round 1
Reviewer 1 Report
The article titled “Is L-glutamate toxic to neurons and thereby contribute to neuronal loss and neurodegeneration? A systematic review” is wonderful review that highlights a leveraged role of excess L-Glutamate in the etiology of neurodegenerative diseases. This reviewer did not find any major concerns with the manuscript. Minor suggestions are as follows:
- In the materials and methods section authors mentioned the key words used for the search, I am wondering, crucial keywords like glutamate and Aβ, glutamate and Tau, glutamate and protein aggregation, may bring few more articles that may relate the role of Glutamate in protein aggregation. For e.g., an article by Kobayashi S [PMID: 31824301], talks about the role of glutamate in tau protein translation.
- In the results section, 3.1.3 authors mentioned about L-Glu enhances AchE activity. In general, AchE activity was reduced in Alzheimers disease. Since the argument made in the section, 3.1.3 is not significant, probably authors could consider removing that section, or probably authors could discuss about thus conundrum in the discussion.
- Figure 2 and Figure 3 beautifully summarizes the overall effect of L-Glu in the brain. Since neurodegenerative diseases feature protein aggregation, if authors could make another small flowchart or figure to highlight probable effects of L-Glu on protein aggregation would be very interesting. For e.g., L-Glu increases neuroinflammation à p38 kinase activity à Tau hyperphosphorylation. Similarly, L-Glu à Calcium levels à Abeta aggregation, something in that line would be very informative for identifying novel drug targets.
- Authors could consider expanding the conclusion little bit.
Author Response
The article titled “Is L-glutamate toxic to neurons and thereby contribute to neuronal loss and neurodegeneration? A systematic review” is wonderful review that highlights a leveraged role of excess L-Glutamate in the etiology of neurodegenerative diseases. This reviewer did not find any major concerns with the manuscript. Minor suggestions are as follows:
We would like to thank Reviewer #1 for the kind and complimentary words regarding our review.
- In the materials and methods section authors mentioned the key words used for the search, I am wondering, crucial keywords like glutamate and Aβ, glutamate and Tau, glutamate and protein aggregation, may bring few more articles that may relate the role of Glutamate in protein aggregation. For e.g., an article by Kobayashi S [PMID: 31824301], talks about the role of glutamate in tau protein translation.
We thank Reviewer #1 for this suggestion as more extensive coverage of elements of Glu-induced protein aggregation should be included in this review, since protein aggregation is a common hallmark of neurodegenerative diseases. Section 3.2.1.10. Protein aggregation within the results section lists the articles that were identified by using the current search terminology, and these are briefly discussed in Section 4.1.1. Cellular and molecular changes within the discussion. We have not expanded the general search terminology further since this review has already captured 71 papers, but an expansion of the details and references regarding the possible mechanisms by which Glu could trigger protein aggregations that typify neurodegenerative diseases is an excellent and welcome suggestion, and one that will enhance the review. Furthermore, Reviewer #1 makes the further excellent suggestion (see below) that this could form the basis of an additional informative Figure. We have therefore performed a separate hand-search of relevant articles that cover Glu-induced protein aggregations and collated these papers. We have expanded the discussion of the revised manuscript to cover these mechanisms and included a new Figure to the revised manuscript. The following text has been included in the revised manuscript:
Collectively, NDDs are characterized histopathologically by the accumulation of extracellular, cytosolic, or nuclear protein oligomers and fibrils, and the formation of these is influenced by an array of protein post-translational modifications (PTMs) [127]. For AD, the accumulation of extracellular Aβ peptide and intracellular hyperphosphorylated tau are thought to be toxic and contribute to neurodegeneration. L-Glu application triggered a significant increase of Aβ (1-42) accumulation in the brain tissue of rats [33,89,91], and increased levels of Aβ (1-40 and 1-42) were also observed in three-month-old rats after neonate L-Glu administration [128]. L-Glu induced a non-significant increase in tau phosphorylation in vitro [52], but neonatal exposure to MSG increased tau phosphorylation in approximately three-month-old rats [129], and three- and six-month-old mice [34,130]. L-Glu also stimulated increased tau translation [131].
Aberrant and neurotoxic protein aggregation could also be increased in response to other molecular mechanisms induced by L-Glu, including protein damage by redox stress such as oxidation of thiol groups and amino acids, and increased PCC [132], increased protein nitration, and altered levels of phosphorylation (Figure 3).
- In the results section, 3.1.3 authors mentioned about L-Glu enhances AchE activity. In general, AchE activity was reduced in Alzheimers disease. Since the argument made in the section, 3.1.3 is not significant, probably authors could consider removing that section, or probably authors could discuss about thus conundrum in the discussion.
We thank Reviewer #2 for raising this point and concur, the experimental evidence of impact of L-Glu on AChE activity is limited. There is only a single in vitro study referred to in Section 3.1.3. that details the exposure to an L-Glu concentration of 100 mM for 3 hours increased AChE activity [62]. There were four in vivo studies with results covered in Section 3.2.1.3. These studies suggest that at relatively high doses of L-Glu (17.5 mg/Kg, p.o [106] or 100 mg/Kg, p.o) [89], AChE was reduced, whereas at a lower dose of L-Glu (6 mg/Kg, p.o) AChE activity was not affected [106], and at even lower doses of L-Glu at 4 mg/Kg (p.o) or 5 mg/Kg (s.c) AChE activity was increased [87,88]. Hence, collectively higher pathological dosing with L-Glu may result in reduced AChE activity (in keeping with the reduced AChE activity observed for AD), whereas lower doses of L-Glu are stimulatory for AChE activity, although dosing regimens were not uniform between studies. We appreciate that an overall conclusion that L-Glu increases AChE activity is not a valid one, and therefore have amended the wording of the subheading of Section 3.2.1.3 to L-Glu administration influences acetylcholinesterase (AChE) activity. We have also adjusted the wording of the discussion section to highlight that the results are equivocal and could reflect the L-Glu concentration applied at dosing and the dosing regimen and time between dosing and analysis. We have also amended Figure 3 to show that AChE activity may increase or decrease (depending on the study).
- Figure 2 and Figure 3 beautifully summarizes the overall effect of L-Glu in the brain. Since neurodegenerative diseases feature protein aggregation, if authors could make another small flowchart or figure to highlight probable effects of L-Glu on protein aggregation would be very interesting. For e.g., L-Glu increases neuroinflammation à p38 kinase activity à Tau hyperphosphorylation. Similarly, L-Glu à Calcium levels à Abeta aggregation, something in that line would be very informative for identifying novel drug targets.
As detailed above, this is an excellent suggestion and we have produced a new Figure 3 that provides details of possible mechanisms of L-Glu-induced protein aggregation.
- Authors could consider expanding the conclusion little bit.
We appreciate that the conclusion is only a few lines which in part reflects the lengthy preceding limitations section. We have therefore renamed this section Summary and Conclusions and provided more text that summarizes the findings more completely as well as providing some concluding statements.
Reviewer 2 Report
This is a timely review focused on L glutamate toxicity to neurons and neurodegenative diseases. the authors clearly provide systematic review on different aspects of L glutamate. I have minor suggestion. it is better to put references at the end of column and remove level of significant. I recommend to publish this review paper.
Author Response
- We are grateful to Reviewer #2 for the supportive comments regarding the publication of this review. The minor suggestions are useful regarding the presentation of the tabularized data. We appreciate that the authors can be listed in the last column but because there will be a direct link applied to the author citation number and the reference list in the finalized (published) version, we think it will be easier for readers to directly access the paper and then assimilate the data and cross-reference it with our table if the authors are listed in the first column. Although not always the case, other systematic style reviews often list the authors in the first column, hence we have followed the general convention.
- We have included the levels of significance in the results tables as these do provide insight into the statistical validation of the studies and provide a point of reference as to whether the studies were qualitative or quantitative. This also highlights the variability between studies as for some only qualitative data was undertaken, in others, no quantitative values were measured (listed as not determined in tables), while some studies set a single threshold of <0.05, while others quantified lower numerical values corresponding to probabilities of 1%, 0.1% etc. Thus, we believe it is useful for the reader to have an appreciation of the levels of significance (if determined) for each of the studies.